# Towards Neuron Attributions in Multimodal Large Language Models

**Junfeng Fang, Zongze Bi, Ruipeng Wang, Houcheng Jiang, Yuan Gao, Kun Wang**[*]
University of Science and Technology of China
{fjf,zacb2018,wrp20021021,janghc,gaoy,wk520529}@mail.ustc.edu.cn

**An Zhang**
National University of Singapore
an_zhang@nus.edu.sg

**Jie Shi**
Huawei
shi.jie1@huawei.com

**Xiang Wang**[*]
University of Science and Technology of China
xiangwang1223@gmail.com

**Tat-Seng Chua**
National University of Singapore
dcscts@nus.edu.sg

## Abstract

As Large Language Models (LLMs) demonstrate impressive capabilities, demystifying their internal mechanisms becomes increasingly vital. Neuron attribution, which attributes LLM outputs to specific neurons to reveal the semantic properties they learn, has emerged as a key interpretability approach. However, while neuron attribution has made significant progress in deciphering text-only LLMs, its application to Multimodal LLMs (MLLMs) remains less explored. To address this gap, we propose a novel **N**euron **A**ttribution method tailored for **M**LLMs, termed **NAM**. Specifically, NAM not only reveals the modality-specific semantic knowledge learned by neurons within MLLMs, but also highlights several intriguing properties of neurons, such as cross-modal invariance and semantic sensitivity. These properties collectively elucidate the inner workings mechanism of MLLMs, providing a deeper understanding of how MLLMs process and generate multi-modal content. Through theoretical analysis and empirical validation, we demonstrate the efficacy of NAM and the valuable insights it offers. Furthermore, leveraging NAM, we introduce a multi-modal knowledge editing paradigm, underscoring the practical significance of our approach for downstream applications of MLLMs. Our code is available at https://github.com/littlelittlenine/NAM_1.

## 1   Introduction

As Large Language Models (LLMs) demonstrate impressive capabilities [1, 2, 3, 4], demystifying their internal mechanisms becomes increasingly vital, particularly in applications emphasizing fairness, trust, and ethical decision-making [5, 6, 7]. To interpret LLMs, "neuron attribution" stands out as a pivotal approach. This method involves attributing text outputs to individual model components (*e.g.,* neurons and hidden layers) to reveal the knowledge and linguistic properties they learn [5, 8, 9, 10, 11, 12]. Such insights not only facilitate tasks like model editing and pruning [13, 14, 15], but also offer a deeper understanding of how LLMs internalize knowledge. For instance, leading neuron-attribution studies [16, 17, 13, 14, 18, 19] suggest that this capability of internalization may predominantly originate from their Feedforward Neural Networks (FFNs).

---

[*]Corresponding authors.

38th Conference on Neural Information Processing Systems (NeurIPS 2024).

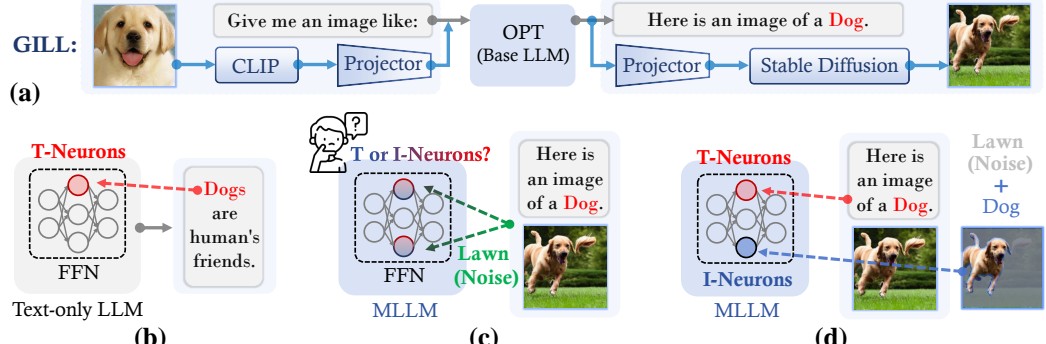

Figure 1: Illustration of neuron attribution methods for interpreting LLMs. (a) The paradigm of GILL; (b) current attribution methods tailored for text-only LLMs; (c) the challenges of extending current attribution methods to MLLMs; (d) the paradigm of our NAM. Best viewed in color.

Recently, the rapid development of multi-modal large language models (MLLMs) [20, 21, 22, 23, 24] is sparking the interest of interpretability [18, 19]. Here we focus on the MLLMs that can perceive, generate texts and images simultaneously. Scrutinizing prior studies, we can summarize two common components beyond the LLM base: image encoding and generating modules. Specifically, the image encoding module projects the input image into the representation space of the base LLM; hereafter, the image generating module generates image outputs conditioned on the representations given by the base LLM. Take GILL [25] as an example. As shown in Figure 1 (a), it hires OPT [26] as the LLM base, CLIP Vit-L [27] with a cross-modal projector as the image encoding module, and Stable Diffusion [28] with another projector as the image generating module.

While this expanded capacity endows GILL with versatility suitable for a variety of downstream tasks, it concurrently presents challenges for interpretation, particularly concerning neuron attributions. Specifically, we outline these challenges through three progressive points:

- **Source of Attribution: Semantic Noise.** As shown in Figure 1 (b), for semantics like dog, current methods typically attribute the output to neurons directly in text-only LLMs [16, 5, 13]. However, in MLLMs, attributing the entire generated image to neurons directly might result in inaccuracies. As shown in Figure 1 (c), when GILL is tasked with drawing a dog, the generated image might contain other semantic elements like lawn, introducing noise and distorting attribution.
- **Process of Attribution: Inefficiency.** Leading attribution methods typically rely on gradients [16, 19] or causal effects [13, 14], requiring extensive forward/backward propagation processes, which are inherently time-consuming and storage-intensive. The added complexity of encoding and generation modules in MLLMs further exacerbates this challenge.
- **Results of Attribution: Intermingled Neurons.** In text-only LLMs, attributing the concept dog involves identifying neurons crucial for outputting the word "dog", termed **T-neurons**. In contrast, MLLMs also require identifying neurons crucial for image generation, called **I-neurons**. As illustrated in Figure 1 (c), the distribution of T-neurons and I-neurons differs for the same concept, leading to conflicting results that complicate further analysis and applications.

In sight of this, we introduce a new **N**euron **A**ttribution paradigm tailored for **M**LLMs, termed **NAM**, to reveal the modality-specific semantic properties learned by neurons within the FFN layers. Specifically, to address the above challenges, NAM undertakes the following efforts:

- **Image Segmentation for Semantic Noise:** As shown in Figure 1 (d), NAM employs the image segmentation model to distinguish regions containing the target semantics from other noisy semantic areas, and attributes these regions to the neurons, rather than the entire image, to ensure accuracy.
- **Activation-based Scores for Inefficiency:** Drawing inspiration from prior studies on neuron activations [18, 5], NAM introduces a new attribution score that relies on neuron activations, eliminating the need for additional forward/backward propagation or gradient calculations.
- **Modality Decoupling for Intermingled Neurons:** NAM assigns modality-specific attribution scores to neurons to prevent cross-modal disturbances during attributions. This paradigm facilitates the decoupling analysis of T-neurons and I-neurons, as depicted in Figure 1 (d).

Furthermore, based on the empirical results of NAM, we reveal several intriguing neuron properties within MLLMs. These properties collectively elucidate the inner workings of MLLMs, enhancing our

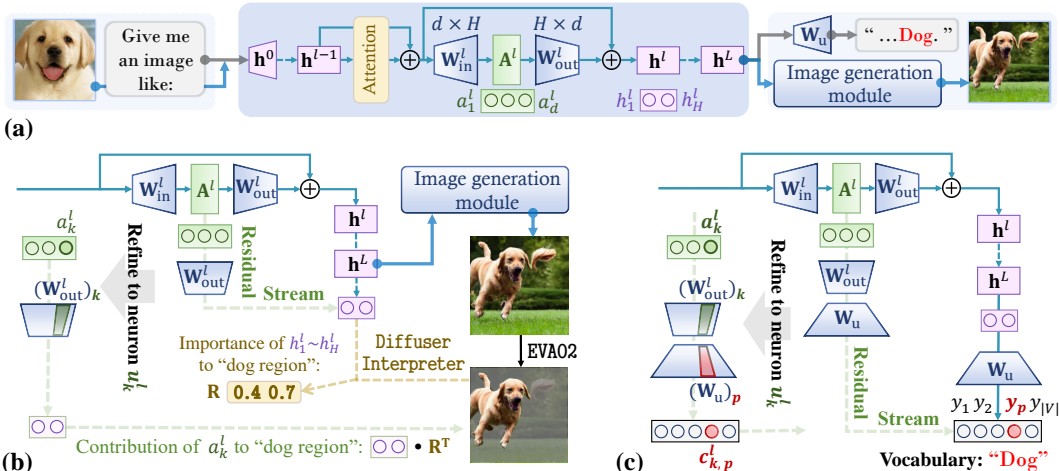

Figure 2: Illustration of neuron attribution methods for interpreting LLMs. (a) The paradigm of current attribution methods tailored for text-only LLMs. (b) The challenges of extending current attribution methods to MLLMs. (c) The paradigm of our NAM.

understanding of their capacity to process and generate multi-modal content. Interestingly, among these insights, a pivotal finding exhibits that when generating multi-model content for the same semantics (*e.g.,* the word "dog" & an image of a dog), the crucial neurons (*i.e.,* T-Neurons & I-Neurons) are typically not identical. This distinction underscores the complex nature of neurons within MLLMs, and highlights the necessity of neuron attribution across modalities. Additionally, by applying NAM to enhance image editing tasks, we further underscore the significance and potential applications of our NAM for MLLM community.

## 2 Preliminary

**Transformer-Based LLMs.** An autoregressive transformer language model $G : \mathcal{X} \rightarrow \mathcal{Y}$ operates over the vocabulary $V$. It receives a token sequence $\mathbf{x} \in \mathcal{X}$ and generates a probability distribution $\mathbf{y} = [y_1, y_2, ..., y_{|V|}] \in \mathcal{Y}$ to predict the next token [29, 13]. Each token is represented as a series of representations $\mathbf{h}^l \in \mathbb{R}^H$ in $l$-th layer, where $\mathbf{h}^0$ is the embedding of the token in $\mathbf{x}$. The model's final output, $\mathbf{y} = \mathbf{W}_{\mathrm{u}} \left( \mathbf{h}^L \right)$, is derived from the last representation $\mathbf{h}^L = \left[ h_1^L, h_2^L, ..., h_H^L \right]^\top$ in layer $L$ using the unembedding matrix $\mathbf{W}_{\mathrm{u}}$. Figure 2 (a) exhibits a visualization of how $\mathbf{h}^l$ are computed within layer $l$. The representation in each layer results from the combination of the global attention $\mathbf{a}^l$, the local MLP output $\mathbf{m}^l$, and the representation $\mathbf{h}^{l-1}$ from the previous layer. Formally,

$$\mathbf{h}^l = \mathbf{m}^l + \mathbf{h}^{l-1} + \mathbf{a}^l, \ \mathbf{m}^l = \mathbf{W}_{\mathrm{out}}^l \sigma \left( \mathbf{W}_{\mathrm{in}}^l \gamma \left( \mathbf{a}^l + \mathbf{h}^{l-1} \right) \right), \quad (1)$$

where $\mathbf{W}_{\mathrm{in}} \in \mathbb{R}^{d \times H}$ and $\mathbf{W}_{\mathrm{out}} \in \mathbb{R}^{H \times d}$ are the first and second linear layer in FFN with the dimensionality of the FFN's intermediate layer $d$; $\sigma$ and $\gamma$ are rectifying and normalizing nonlinearity. For further background on transformers, we refer to [29]. Additionally, focusing on the the $k$-th neuron $u_k^l$ in the $l$-th FFN layers, we simplify the definition by considering its activation $a_k^l$ as:

$$\mathbf{A}^l = [a_1, a_2, ..., a_d]^\top = \sigma \left( \mathbf{W}_{\mathrm{out}}^l \gamma \left( \mathbf{a}^l + \mathbf{h}^{l-1} \right) \right) \quad (2)$$

**MLLMs.** Here, we take GILL as an example to illustrate a common paradigm of MLLMs. As shown in Figure 2 (a), GILL incorporates the following modifications on the above text-only LLMs: (1) In addition to the textual prompt, the input token sequence $\mathbf{x}$ also includes the encoding of the input image produced by the image encoding module. (2) The representation $\mathbf{h}^L$ of the last hidden layer is utilized as input to the image generation module, facilitating conditional image generation.

## 3 Method

This section delineates the implementation of NAM within MLLMs. Specifically, Section 3.1 and 3.2 introduce the attribution process for image and text outputs, respectively; In Section 3.3, we

explore a significant application of NAM, *i.e.,* editing images generated by MLLMs. To illustrate these processes, we utilize GILL [25], a representative MLLM capable of image generation. For a detailed introduction to GILL, please refer to Appendix B.

## 3.1 Neuron Attribution for Image Generation

We first detail how to attribute the output images to the specific neurons within FFN layers. Retrospecting the challenges highlighted in Introduction, using the MLLM to generate images of a specific concept (*e.g.,* dog) often results in outputs that include extraneous, noisy elements (*e.g.,* lawn). To mitigate the negative effects of semantic noise on neuron attribution, we propose a two-step approach to extract I-Neurons. Specifically, (1) the **first step** focuses on attributing the output of the image generation module (*i.e.,* images) to the input of the image generation module (*i.e.,* last representation $\mathbf{h}^L$), and (2) the **second step** endeavors to attribute the input of the image generation module to the specific neurons. Next, we provide detailed descriptions of these two steps.

### 3.1.1 STEP1: Attribution From Images to Representation $\mathbf{h}^L$

The purpose of this step is to attribute the image to $\mathbf{h}^L \in \mathbb{R}^H$. That is, to identify the contribution of each element in $\mathbf{h}^L$ for image generation. We define these contribution scores as $\mathbf{R} \in \mathbb{R}^H$. As shown in Figure 2 (b), NAM acquires $\mathbf{R}$ by sequentially executing the following processes:

- Given an image generated by prompting MLLM with the semantics dog, NAM first employs the leading segmentation model, EVA02 [30], to identify regions specifically related to dog. This is crucial for minimizing interference from extraneous semantics, such as lawn in the background;
- Subsequently, NAM utilizes the advanced attribution algorithm of the diffusion model, Diffuser-Interpreter [31], to access the relevance of each dimension in the input of the image generation module to the dog region in the generated image.
- Ultimately, by normalizing these relevance scores to $(0, 1)$, we obtain the importance scores $\mathbf{R} = [r_1, r_2, ..., r_H]^\top$ of $\mathbf{h}^L = [h_1, h_2, ..., h_H]^\top$ *w.r.t* the target semantics in the output image.

Due to the established applications of EVA02 and Diffuser-Interpret, we provide detailed introductions in Appendix B. Furthermore, it is worth mentioning that NAM can be transferred to any other modality by utilizing the (1) semantic segmentation algorithms and (2) attribution algorithms of generation modules tailored for other modalities (*e.g.,* audio and video). After obtaining $\mathbf{R}$ by these advanced modality-specific algorithms, the subsequent attribution steps are universal across all transformer-based MLLMs.

### 3.1.2 STEP2: Attribution From Representation $\mathbf{h}^L$ to Neuron $u_k^l$

This step involves attributing the representation $\mathbf{h}^L$ in the last layer to the specific neuron $u_k^l$ within the FFNs of the base LLM. To this end, NAM aims to trace each neuron's contribution to $\mathbf{h}^L$, and identify the neurons with significant contributions as I-Neurons for the semantic of interest. Hence, a fair and efficient contribution scoring method is crucial.

**Direct Contributions through Residual Stream.** Current methods for scoring contributions often rely on gradients, such as the product of gradients and activations [19] or the integration of gradients [16]. However, these methods are computationally intensive, particularly for large-scale models with extensive parameters. In sight of this, drawing inspiration from prior studies on neuron activation [18, 19], we first introduce a new attribution score that relies on the neuron activation $a_k^l$. Specifically, we first try to disassemble and deduce the generation procedure of $\mathbf{h}^L$ by expanding $\mathbf{h}^L$ as follows:

$$
\begin{aligned}
\mathbf{h}^L = \mathbf{m}^L + \mathbf{h}^{L-1} + \mathbf{a}^L &= \sum_{l=1}^{L} \mathbf{m}^l + \mathbf{h}^0 + \sum_{l=1}^{L} \mathbf{a}^l \\
&= \sum_{l=1}^{L} \mathbf{W}_{\text{out}}^l \mathbf{A}^l + \mathbf{h}^0 + \sum_{l=1}^{L} \mathbf{a}^l = \sum_{l=1}^{L} \sum_{k=1}^{d} a_k^l (\mathbf{W}_{\text{out}}^l)_k + \mathbf{h}^0 + \sum_{l=1}^{L} \mathbf{a}^l,
\end{aligned}
\tag{3}
$$

where $(\mathbf{W}_{\text{out}}^l)_k \in \mathbb{R}^H$ is the $k$-th column of the weight matrix $\mathbf{W}_{\text{out}}^l$ corresponding to the index of neuron $u_k^l$, as shown in Figure 2 (b). Note that the first term of Equation (3) reflects the direct contribution of the neuron $u_k^l$ to the last representation $\mathbf{h}^L$, *i.e.,* the contribution through the residual stream [19] of the base LLM.

Hence, we employ $a_k^l(\mathbf{W}_{\text{out}}^l)_k$ as the indicator for the neuron $u_k^l$'s contribution to $\mathbf{h}^L$.

Furthermore, by integrating $\mathbf{R}$ in Section 3.1.1, we can establish a complete attribution pipeline (*i.e.,* targeted semantic region in generated image $\Rightarrow$ representation $\mathbf{h}^L \Rightarrow$ neuron $u_k^l$). Recall that $\mathbf{R}$ has assigned the contribution for each dimension in $\mathbf{h}^L$, the neuron $u_k^l$'s contribution $s_k^l$ can be defined as $a_k^l(\mathbf{W}_{\text{out}}^l)_k$ weighted by elements in $\mathbf{R}$. That is, $s_k^l = a_k^l(\mathbf{W}_{\text{out}}^l)_k\mathbf{R}^\top$, as illustrated in Figure 2 (b).

**Contribution Score Considering Indirect Influence.** While $s_k^l$ quantifies the neuron $u_k^l$'s direct contribution to $\mathbf{h}^L$ through the residual stream, it does not account for all influential factors. Specifically, it overlooks the indirect contributions that neurons make through the attention mechanisms within subsequent FFN layers. Supporting evidence exhibited in Appendix B verifies that this oversight might lead to a bias. To address this issue, and in line with our objective to eschew complex computations like gradient, we implement a heuristic optimization of the current indicator $s_k^l$. Specifically, we employ the relative magnitude of neuron activation as another indicator to identify neurons that may have a significant indirect contribution to $\mathbf{h}^L$ – contribution which $s_k^l$ might overlook.

Furthermore, considering the computation of $s_k^l$ already incorporates $a_k^l$, the contributions reflected by these two indicators may overlap. To prevent redundancy from summing or multiplying these two metrics, we utilize the maximum function in our final score design:

$$\hat{s}_k^l = \max\Big\{\frac{e^{s_k^l}}{\sum_{l=1}^{L}\sum_{k=1}^{d} e^{s_k^l}}, \frac{e^{a_k^l}}{\sum_{l=1}^{L}\sum_{k=1}^{d} e^{a_k^l}}\Big\}, \tag{4}$$

where the normalization operation ensures fair competition between $s_k^l$ and $a_k^l$. Note that our experiments in Section 4 have shown that this combined scoring approach is more effective than utilizing $s_k^l$ or $a_k^l$ alone, verifying their complementary nature. By computing contributions following Equation (4) for various semantics across all layers, NAM identifies neurons that consistently demonstrate the highest contributions to the generated images. These neurons, distinguished by their significant roles, are designated as I-neurons responsible for targeted semantics in MLLMs.

## 3.2 Neuron Attribution for Text Generation

We then focus on how to acquire the neuron's contribution to the text outputs. Similar to the derivation process of contribution score for image output, here we first focus on the contribution for output $\mathbf{y}$ through the residual stream. Specifically, we expand $\mathbf{h}^L$ as follows:

$$\mathbf{y} = \mathbf{W}_{\text{u}}\big(\mathbf{m}^L + \mathbf{h}^{L-1} + \mathbf{a}^L\big) = \sum_{l=1}^{L} \mathbf{W}_{\text{u}}\mathbf{W}_{\text{out}}^l \mathbf{A}^l + \mathbf{W}_{\text{u}}(\mathbf{h}^0 + \sum_{l=1}^{L} \mathbf{a}^l). \tag{5}$$

According to a commonly used assumption for analyzing the internal mechanisms of LLMs, representations at any layer within the language models can be transformed into a distribution over the token vocabulary $V$ using the output embeddings [18, 19, 32, 16, 13, 14]. Hence, $\mathbf{W}_{\text{u}}\mathbf{W}_{\text{out}}^l \in \mathbb{R}^{|V|\times H}$ can be considered as the new unembedding matrix at the end of the residual stream, and $\mathbf{A}^l$ contributes to the model output distribution $\mathbf{y}$ over the vocabulary through $\mathbf{W}_{\text{u}}\mathbf{W}_{\text{out}}^l\mathbf{A}^l$, as shown in Figure 2 (c).

**Refined Contribution of Individual Neuron.** We further disassemble Equation (5) to refine individual neuron $u_k^l$'s contribution to the output word. Specifically, denoting $p$ as the index of the word "dog" in the vocabulary $V$, we have:

$$y_p = (\mathbf{W}_{\text{u}})_p\mathbf{W}_{\text{out}}^l\mathbf{A}^l + (\mathbf{W}_{\text{u}})_p(\mathbf{h}^0 + \sum_{l=1}^{L} \mathbf{a}^l) = \sum_{k=1}^{d} a_k^l(\mathbf{W}_{\text{u}})_p(\mathbf{W}_{\text{out}}^l)_k + (\mathbf{W}_{\text{u}})_p(\mathbf{h}^0 + \sum_{l=1}^{L} \mathbf{a}^l), \tag{6}$$

where $(\mathbf{W}_{\text{u}})_p \in \mathbb{R}^{1\times d}$ is the $p$-th row of $\mathbf{W}_{\text{u}}$. According to Equation (6), the neuron $u_k^l$'s contribution $c_{k,p}^l$ to the $p$-th word "dog" on vocabulary can be obtained by $c_{k,p}^l = a_k^l(\mathbf{W}_{\text{u}})_p(\mathbf{W}_{\text{out}}^l)_k$, as illustrated in Figure 2 (c). Furthermore, we would like to encourage the semantic specificity of the identified crucial neurons – that is, only preserving a single semantic concept with the maximum contribution, while discarding other semantics. Formally, for the semantics of $p$-th word on vocabulary, the neuron $u_k^l$'s contribution $s_k^l$ can be defined as:

$$p^* = \arg\max_{p} c_{k,p}^l, \quad s_k^l = \begin{cases} c_{k,p}^l & \text{if } p = p^*, \\ 0 & \text{otherwise.} \end{cases} \tag{7}$$

By substituting $s_k^l$ to Equation (4), we can acquire the final attribution score of neuron $u_k^l$ for text output. The neurons that consistently exhibit the highest contributions are then designated as T-neurons.

### 3.3 Image Editing Enhanced by NAM

Knowledge editing methods based on neuron attribution have already been explored for text outputs [13, 14, 33, 34]. Here, we focus on how to leverage attribution results to facilitate knowledge editing of images. This objective requires replacing some semantics (*e.g.,* dog) with another semantics (*e.g.,* cat). To this end, we leverage the I-Neurons identified by NAM for image editing through a straightforward, training-free approach. Specifically, we first construct the set of I-Neurons, $\mathcal{U}$, for the semantics like dog. Then, we collect the positions $(l, k)$ of the neurons $u_k^l$ in $\mathcal{U}$, and construct the set of these position indices, $\mathcal{I}$. For $(l, k) \in \mathcal{I}$, we add a perturbation $\Delta(\mathbf{W}_{\text{out}}^l)_k$ to $(\mathbf{W}_{\text{out}}^l)_k$ following:

$$\mathcal{W} = \{\Delta(\mathbf{W}_{\text{out}}^l)_k \text{ for } (l, k) \in \mathcal{I}\}$$

$$\Delta(\mathbf{W}_{\text{out}}^l)_k = \arg\min_{\mathcal{W}} || \sum_{(l,k)\in\mathcal{I}} a_k^l \Delta(\mathbf{W}_{\text{out}}^l)_k, (\hat{\mathbf{h}}^L - \mathbf{h}^L)^\top ||_2 + \tau || \sum_{(l,k)\in\mathcal{I}} \Delta(\mathbf{W}_{\text{out}}^l)_k \cdot \mathbf{1}^\top ||_2, \quad (8)$$

where $\mathbf{h}^L$ and $\hat{\mathbf{h}}^L$ is the last representation of the base LLM when generating the image of dog and cat, respectively; $\tau$ serves as a trade-off parameter. In Equation (8), the first term aims to facilitate a shift in the image generation module's input from $\mathbf{h}^L$ to $\hat{\mathbf{h}}^L$, while the second term is the $\ell_2$ norm constraint for preventing drastic edits that might affect images containing other semantics. According to this method, NAM can be utilized to enable simple and efficient images editing, underscoring the significance and potential applications of the NAM for MLLMs[2].

## 4 Experiment

In this section, we aim to validate the effectiveness of NAM from three aspects:

- What is the distribution of T/I-Neurons identified by NAM?
- What properties do the T/I-neurons identified by NAM have? How to verify that the T/I-neurons identified by NAM are more critical compared to the neurons identified by baseline methods?
- Can the T/I-Neurons identified by NAM facilitate the image editing within MLLMs?

### 4.1 Investigation Setup

**Target Models & Datasets.** Our research focuses on GILL [25] and NExT-GPT [35], two representative MLLMs with the capability of image generation. All experiments are conducted on the Common Objects in Context (COCO) [36], a large-scale object detection, segmentation, and captioning dataset including 80 object categories and five captions per image to conduct our experiments. Due to space limitations, we only present the experimental results on GILL in this section. The remains and the detailed implementations, such as the setting of hyper-parameters, can be found in Appendix B.

**Baselines.** We collect five advanced neuron attribution methods across three categories (gradient-, activation-, and causality-based attribution). Specifically, their abbreviations and the attribution scores they employ are: (1) AcT: neuron activation [5]; (2) AcU: The product of activation and the unembedding matrix, focused on the dimension corresponding to the output word [18]; (3) GraD: The gradient of the output dimension corresponding to the output word *w.r.t* activation. (4) GraT: The product of the gradients and activation [19]; (5) GraI: The integral of the gradients [16]; (6) CE: The causal effect of activation on outputs [13, 14]. See detailed description in Appendix B.1. For NAM and baselines stand and their role in rich literature, please refer to Appendix A (*i.e.,* Related Work).

### 4.2 RQ1: Distribution of T/I-Neurons

We first focus on the distribution of T/I-neurons identified by NAM. Specifically, we randomly select 1000 images from the COCO dataset. Then, we feed each image to GILL individually and instruct GILL to generate a similar image. The distributions of T/I-neurons are exhibited in Figures 3 (a) and

---

[2]The extension of NAM to broader scenarios will be detailed in Appendix C.

Table 1: The semantics of T/I-neurons with the highest attribution scores identified by different attribution methods, when the output of MLLMs containing the targeted semantics. (L$l$,U$k$) denotes the $k$-th neuron at layer $l$. For each method, we report semantics with top-4 probabilities.

| Output | Segmentation | Category | Method | Location | Semantics |
|---|---|---|---|---|---|
| *"A girl is riding a* **horse**."  |  | T-Neurons | Grad | L28.U4786 | {'dog', 'shark', 'cat', 'bird'} |
| | | | AcT | L30.U13868 | {'animals', 'animal', 'Animal', 'Animals'} |
| | | | CE | L27.U14262 | {'vehicles', 'trucks', 'cars', 'boats'} |
| | | | **NAM** | **L23.U5318** | {**'horses', 'horse', 'Horses', 'Horses'**} |
| | | I-Neurons | Grad | L29.U14374 | {'farming', 'farm', 'farms', 'ag'} |
| | | | AcT | L26.U12957 | {'animal', 'animals', 'veterin', 'veterinary'} |
| | | | CE | L28.U1208 | {'Kinnikuman', 'cffff', 'Nanto', 'Vaults'} |
| | | | **NAM** | **L23.U5318** | {**'horses', 'horse', 'Horses', 'Horses'**} |
| *"A* **dog** *is running on the lawn."*  |  | T-Neurons | Grad | L28.U12056 | {'child', 'Child', 'children', 'male'} |
| | | | AcT | L24.U12845 | {'dogs', 'dog', 'Dog', 'canine'} |
| | | | CE | L25.U3655 | {'those', 'Those', 'that', 'this'} |
| | | | **NAM** | **L24.U10710** | {**'dogs', 'Dog', 'Dogs', 'pets'**} |
| | | I-Neurons | Grad | L26.U1135 | {'adopt', 'pet', 'adopting', 'adoption'} |
| | | | AcT | L30.U13868 | {'animals', 'animal', 'Animal', 'Animals'} |
| | | | CE | L31.U1135 | {'weeds', 'chickens', 'compost', 'trash'} |
| | | | **NAM** | **L24.U12845** | {**'dogs', 'dog', 'Dog', 'canine'**} |
| *"A small* **ship** *on the sea."*  |  | T-Neurons | Grad | L26.U3972 | {'diving', 'digging', 'dred', 'drilling'} |
| | | | AcT | L28.U11438 | {'boat', 'car', 'phone', 'vehicle'} |
| | | | CE | L30.U3335 | {'inar', 'set', 'Set', 'cam'} |
| | | | **NAM** | **L30.U2503** | {**'ship', 'ships', 'Ship', 'shipping'**} |
| | | I-Neurons | Grad | L27.U8984 | {'bush', 'tree', 'brush', 'shr'} |
| | | | AcT | L25.U2539 | {'swim', 'swimming', 'Swim', 'underwater'} |
| | | | CE | L25.U5113 | {'bronze', 'sign', 'box', 'SIGN'} |
| | | | **NAM** | **L28.U10626** | {**'ship', 'ships', 'sea', 'ocean'**} |

Figure 3: Distribution of (a) I-neurons, (b) T-neurons, and (c) intersection and subset of I-neurons and T-neurons per layer identified by NAM, chosen by different number of neurons with top scores on average. Best viewed in color.

(b), while Figure 3 (c) illustrates the distribution of intersections of T- and I-neurons. These results demonstrate the following observations:

**Observation 1:** Within MLLMs, the crucial neurons for the text and image output containing specific semantics predominantly occur in the middle and high layers of the base LLM. Note that this observation is consistent with the previous works involving neuron attributions within LLMs [5, 13, 18]. Additionally, the similar distribution of T and I neurons suggests that the formation time of semantic concepts across different modalities in MLLMs may be consistent.

**Observation 2:** Figure 3 (c) reveals a partial overlap between T and I neurons. However, it also pronounces distinctions between them. This finding substantiates the claim presented in the Introduction of this paper: even for the same semantics, critical neurons are modality-specific within MLLMs.

### 4.3 RQ2: Properties of T/I-Neurons & Effectiveness of NAM

We then explore the following properties of T/I-neurons through comprehensive quantitative and qualitative experiments: **Semantic Relevance**, **Cross-Sample Invariance**, and **Concept Specificity**. Additionally, by comparing these properties with those of neurons identified by various baselines, we validate the effectiveness of our NAM. Note that we only present the results of the best-performing baseline for each class, and remains are shown in Appendix B.

#### 4.3.1 Semantic Relevance

Following previous studies [19, 18], we treat the unembedding matrix and the second linear layer matrix in the FFN as a projection from neuron activation to the probability distributions of the vocabulary. Based on this, words with the average highest probability can be regarded as the relevant

Table 2: Consistency between the neuron's semantics and the images/captions. *Grad.*, *Act.* and *Ca.* denote gradient-, activation- and causality-based methods, respectively. ‡ and ⋆ represent the CLIPScore *w.r.t* input and output images. We use background ▨ highlights the best performance.

| | Class | | | T-Neurons | | | | |
|---|---|---|---|---|---|---|---|---|
| **Method** | **Grad.** | **Act.** | **Ca.** | **‡CLipScore** | **⋆CLipScore** | **BERTScore** | **MoveScore** | **BLEURT** |
| CE | | | ✓ | $0.264_{\pm0.015}$ | $0.251_{\pm0.022}$ | $0.273_{\pm0.029}$ | $0.257_{\pm0.019}$ | $0.040_{\pm0.005}$ |
| GraI | ✓ | ✓ | | $0.239_{\pm0.018}$ | $0.244_{\pm0.020}$ | $0.276_{\pm0.032}$ | $0.296_{\pm0.030}$ | $0.039_{\pm0.005}$ |
| GraD | ✓ | | | $0.378_{\pm0.047}$ | $0.396_{\pm0.032}$ | $0.457_{\pm0.027}$ | $0.436_{\pm0.030}$ | $0.064_{\pm0.008}$ |
| GraT | ✓ | ✓ | | $0.425_{\pm0.040}$ | $0.422_{\pm0.029}$ | $0.486_{\pm0.018}$ | $0.477_{\pm0.036}$ | $0.072_{\pm0.009}$ |
| AcT | | ✓ | | $0.556_{\pm0.037}$ | $0.594_{\pm0.046}$ | $0.624_{\pm0.057}$ | $0.653_{\pm0.054}$ | $0.139_{\pm0.013}$ |
| AcU | | ✓ | | $0.543_{\pm0.051}$ | $0.624_{\pm0.057}$ | $0.618_{\pm0.054}$ | $0.609_{\pm0.038}$ | $0.135_{\pm0.014}$ |
| NAM | | ✓ | | $0.562_{\pm0.054}$ | $0.637_{\pm0.047}$ | $0.640_{\pm0.039}$ | $0.657_{\pm0.048}$ | $0.148_{\pm0.013}$ |
| | Class | | | I-Neurons | | | | |
| **Method** | **Grad.** | **Act.** | **Ca.** | **‡CLipScore** | **⋆CLipScore** | **BERTScore** | **MoveScore** | **BLEURT** |
| CE | | | ✓ | $0.228_{\pm0.017}$ | $0.219_{\pm0.026}$ | $0.245_{\pm0.033}$ | $0.250_{\pm0.021}$ | $0.044_{\pm0.003}$ |
| GraI | ✓ | ✓ | | $0.230_{\pm0.021}$ | $0.235_{\pm0.021}$ | $0.259_{\pm0.027}$ | $0.278_{\pm0.037}$ | $0.035_{\pm0.004}$ |
| GraD | ✓ | | | $0.370_{\pm0.042}$ | $0.377_{\pm0.026}$ | $0.432_{\pm0.040}$ | $0.409_{\pm0.041}$ | $0.058_{\pm0.006}$ |
| GraT | ✓ | ✓ | | $0.432_{\pm0.038}$ | $0.394_{\pm0.032}$ | $0.453_{\pm0.042}$ | $0.437_{\pm0.038}$ | $0.068_{\pm0.008}$ |
| AcT | | ✓ | | $0.547_{\pm0.043}$ | $0.580_{\pm0.051}$ | $0.597_{\pm0.056}$ | $0.623_{\pm0.061}$ | $0.128_{\pm0.013}$ |
| AcU | | ✓ | | $0.501_{\pm0.061}$ | $0.601_{\pm0.054}$ | $0.559_{\pm0.039}$ | $0.548_{\pm0.052}$ | $0.137_{\pm0.013}$ |
| NAM | | ✓ | | $0.558_{\pm0.053}$ | $0.613_{\pm0.052}$ | $0.611_{\pm0.048}$ | $0.630_{\pm0.056}$ | $0.144_{\pm0.014}$ |

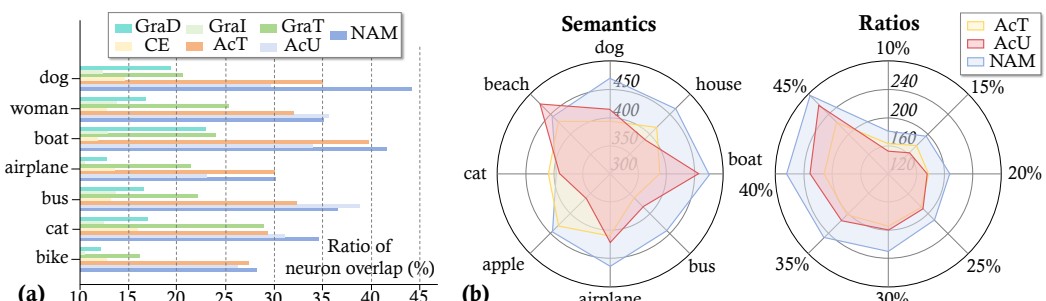

Figure 4: Cross-sample invariance and semantic specificity of T/I-neurons. (a) exhibits invariance by calculating the average ratio between T/I-neurons' subset and intersection across different text/image output; (b) quantifies specificity by showing: 1. the number of neurons crucial for specific semantics solely and 2. the average number of neurons whose probability of being crucial to other semantics is lower than a certain value. Best viewed in color.

semantics of neurons, as shown in Table 1. Note that the results of AcU are not exhibited since its attribution score is exactly this probability. Furthermore, to explore the semantic relevance of the neurons quantitatively, we calculate the consistency between the semantics of neurons and the input/output images employing CLIPScore [37]. BERTScore [38], MoverScore [39], and BLEURT [40] are also employed to quantify their consistency with (1) input image's caption provided by the dataset and (2) output image's caption given by GPT [1]. Table 2 exhibits the average quantified results. According to Table 1 and 2, we have the following observation:

**Observation 3:** The semantics of T/I-neurons identified by NAM align more closely with the input/output images and their captions, while the other attribution methods typically identify the neurons that are hardly correlated with the targeted semantics. The quantitative results share a similar tendency, confirming the high semantic relevance of T/I-neurons and the effectiveness of our NAM.

### 4.3.2 Cross-sample Invariance

For different text/image outputs containing the same semantics, the T/I-neurons identified by the attribution methods shall be consistent. To quantify this consistency, we instruct GILL to describe and generate images for the same semantics ten times, and collect the set of T/I-neurons each time. We then calculate the proportion of neurons that appeared in all ten sets as the quantification of

cross-sample invarance. The average invarance Figure 4 (a) presents the average invarance for different concepts. Specifically,

**Observation 4:** NAM outperforms all baselines by an average of 16.83% *w.r.t* cross-sample invariance across all semantics. This demonstrates that NAM extracts the critical neurons for the targeted semantics across samples, effectively filtering out the neurons sensitive to sample-specific noise.

### 4.3.3 Semantic Specificity

Neurons that are crucial for specific semantics should not be indiscriminately crucial across others. Therefore, we study the neuron's specificity in this part. We identify the Top-500 T/I-neurons for the specific semantics. Then, we show (1) the number of neurons that are crucial for specific semantics solely and (2) the average number of neurons whose probability of being crucial to other semantics is lower than $\kappa \in \{10\%, 15\%, \ldots, 45\%\}$ in Figure 4 (b). The results of the three best-performing methods are exhibited here. These results highlight that:

**Observation 5:** The T/I-neurons identified by our NAM are specialized and not commonly sensitive across different semantics, verifying their specificity across the semantics.

| Pre-editing | Post-editing | Method | Value ($\downarrow$) |
|---|---|---|---|
| *"boy"* | *"girl"* | CE | 0.726 |
| | | GraI | 0.625 |
| | | GraD | 0.471 |
| | | GraT | 0.502 |
| | | AcT | 0.390 |
| | | AcU | 0.362 |
| | | **NAM** | **0.350** |
| *"dog"* | *"cat"* | CE | 0.601 |
| | | GraI | 0.540 |
| | | GraD | 0.458 |
| | | GraT | 0.321 |
| | | AcT | 0.259 |
| | | AcU | 0.288 |
| | | **NAM** | **0.255** |
| *"cantaloupe"* | *"apple"* | CE | 0.493 |
| | | GraI | 0.360 |
| | | GraD | 0.357 |
| | | GraT | 0.401 |
| | | AcT | 0.328 |
| | | AcU | 0.324 |
| | | **NAM** | **0.316** |

Table 3: Results of image editing. The Value represents the $\ell_2$ norm of the perturbation added to the I-neurons, demonstrating that NAM necessitates minimal perturbations for the editing.

## 4.4 RQ3: Image Editing Enhanced by NAM

Lastly, we aim to edit the images generated by MLLMs through perturbing I-neurons identified by NAM, as outlined in Section 3.3. Table 3 exhibits the pre- and post-editing semantics, the selected images for collecting $\mathbf{h}^L$ and $\hat{\mathbf{h}}^L$, and the magnitude of the perturbations. Furthermore, for fair comparisons, we also perturb I-neurons identified by baselines to achieve similar editing results. According to Table 3 we can find that:

**Observation 6:** NAM-enhanced editing methods can not only replace the original semantics with the target semantics precisely within the outputs of MLLMs, but also necessitate minimal perturbations. Specifically, the perturbation it added is 40.2% less than the baselines on average, and nearly 15% less than the best baseline, underscoring its significance and potential applications for MLLMs.

## 5 Limitations & Future Work

This study provides new insights into interpreting MLLMs, enhancing the understanding of their inner working mechanize. However, while our experiments thoroughly investigated the neuron properties within GILL and NExTGPT, they did not extend to a broader range of models. Additionally, although the proposed attribution method can be transferred to any other modality, as demonstrated in Section 3.1.1, our experiments focused on text and image outputs solely. Looking forward, we plan to incorporate more MLLMs and modalities into our research, and streamline our attribution method to eliminate the reliance on external interpreters. By expanding the scope of our study and refining our method, we aim to uncover more valuable insights that will benefit the MLLM community.

## 6 Conclusion

We propose NAM, a novel neuron attribution method tailored for MLLMs. Specifically, NAM is tailored for multi-modal attribution, revealing the modality-specific semantic properties learned by neurons within the FFN layers.To address the challenges of extending attribution methods from text-

only LLMs to MLLMs, NAM first employs a leading image segmentation model to remove the noisy semantics, then proposes a new attribution score to eliminate the need for additional forward/backward propagation or gradient calculations. Based on NAM, we highlights several intriguing properties of neurons, elucidating the inner workings mechanism of MLLMs.

## Acknowledgments

This research is supported by the National Science and Technology Major Project (2023ZD0121102), and National Natural Science Foundation of China (92270114).

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

# A Related Work

**Multi-modal Large Language Models.** In recent years, MLLMs have made significant progress, continually pushing the boundaries of performance in various downstream tasks [41]. The launch of models like GPT-4 (Visual) and Gemini, with their impressive multimodal (MM) understanding and generation capabilities, has sparked intense research interest in MM-LLMs [20]. In terms of understanding multimodal content, preliminary research often utilizes multimodal encoders like the ViT [42] and CLIP [27] ViT to capture representations, and projectors like Q-Former [43, 44] and P-Former [45] to align these representations with the embedding space of foundational LLMs. This approach covers tasks such as image-text understanding, with representative models including BLIP-2 [21], MiniGPT-4 [23], LLAVA [22], and OpenFlamingo [24].

Recently, the capabilities of MM-LLMs have expanded to support specific modal outputs. These methods align certain embeddings from the foundational LLM with the input space of a well-trained multimodal generator through another projector. This extension includes tasks with image-text outputs as demonstrated by models like GILL [25], Kosmos-2 [46], Emu [47], NExT-GPT [35] and MiniGPT-5 [48], which is the focus of this paper. Recent research endeavors have focused on mimicking human-like any-to-any modality conversion, shedding light on the path to artificial general intelligence [20].

**Interpretability of Pre-trained Transformers.** Demystifying the internal mechanisms of LLMs becomes increasingly vital, particularly in applications emphasizing fairness, trust, and ethical decision-making [49, 50, 51, 52, 53, 54, 55]. In recent years, many works have focused on explaining pre-trained transformers. For instance, [32] regards the FFN as unnormalized Key-Value Memories; [56] presents a conceptual framework where all parameters are interpreted by projecting them into the embedding space; [9] analyzes the FFN updates in the vocabulary space, showing that each update can be decomposed to sub-updates; [17] studies how the model aggregates information about the subject and relation to predict the correct attribute; while [11] localizes the weights and mechanisms used by a language model to memorize and recite entire paragraphs of its training data.

**Neuron Attribution.** Neuron attribution aims to reveal the black box of pre-trained transformers by answering the following questions [7, 57, 6]: (1) What concepts are learned within neurons of the network? (2) Are there neurons that specialize in learning particular concepts? (3) How localized/distributed and redundantly is the knowledge preserved within neurons of the network? To achieve these goals, many recent works have focused on exploring the properties of neurons in LLMs. Specifically, [16] present preliminary studies on how factual knowledge is stored in LLMs by introducing the concept of knowledge neurons; [5] finds the special neurons whose activations on soft prompts are highly predictive of the task labels of inputs, and dub them skill neurons.

For MLLMs, [18] employs the neuron contribution within the residual stream to identify multi-modal neurons in Transformer-based multi-modal LLMs, while [19] claims that image prompts cast into the transformer embedding space do not encode interpretable semantics, and translation between modalities occurs inside the transformer. Additionally, despite the aforementioned methods are all based on activation and gradients to identify target neurons, some recent works focused on knowledge editing have discovered an alternative approach [15, 13]. This involves using causal effect methods to identify key neurons by perturbing neurons and observing changes in the output [14].

For the detailed contribution scores these works employed and their relationship with our NAM, please see Appendix B.1.

# B More Experiments

## B.1 Experimental Settings

**Baseline.** Here we first detail the contribution score of the baseline methods used. We will use the formulas and symbols from the paper to provide a formal explanation. Specifically, for the $p$-th word in Vocabulary $V$ and the $k$-th neuron $u_k^l$ in layer $l$, we have:

- AcT takes the neuron activation as the contribution score following:

$$\hat{s}_k^l = a_l^k. \tag{9}$$

- AcU takes the product of the neuron activation and the unembedding matrix as the contribution score following:

$$\hat{s}_k^l = c_{k,p}^l = a_k^l (\mathbf{W}_u)_p (\mathbf{W}_{\text{out}}^l)_k. \tag{10}$$

- GraD takes the gradient of the output dimension corresponding to the output word *w.r.t* activation as the contribution score following:

$$\hat{s}_k^l = \frac{\partial y_p}{\partial a_l^k}. \tag{11}$$

- GraT takes the product of the gradients and activation as the contribution score following:

$$\hat{s}_k^l = a_l^k \frac{\partial y_p}{\partial a_l^k}. \tag{12}$$

- GraI takes the product of the integral of the gradients and the activation as the contribution score following:

$$\hat{s}_k^l = a_l^k \int_{\alpha=0}^{1} \frac{\partial y_p'}{\partial a_l^k} d\alpha, \tag{13}$$

where $y_p'$ is the $p$-th dimension of the output $\mathbf{y}$ when we fix the activation of neuron $u_k^l$ as $\alpha \cdot a_l^k$. Note that perturbing and performing forward propagation for each neuron multiple times is time-consuming. Therefore, we add the same coefficient $\alpha$ to multiple neurons simultaneously and calculate their integrals to obtain their respective contribution scores.

- CE takes the causal effect of activation on outputs, quantified by the output change when activation is perturbed, as the contribution score following:

$$\hat{s}_k^l = \frac{\partial y_p''}{\partial a_l^k}, \tag{14}$$

where $y_p''$ is the $p$-th dimension of the output $\mathbf{y}$ when we add the activation of neuron $u_k^l$ with a Gaussian Noise. We set the mean of the Gaussian noise to 0 and the variance to the total variance of the neuron activation values in the respective layer. Similar to GraI, for each individual neuron, performing forward propagation for each neuron multiple times is costly. Therefore, we add perturbations to multiple neurons simultaneously and evenly distribute the resulting output changes among them as their respective contribution scores.

Furthermore, it is important to note that the above baselines all attribute text outputs to neurons. Therefore, we have adapted all these baselines to image outputs for comprehensive validation. For example, the attribution score of the baseline *Grad* for text output is the gradient of activation when the output is the word "dog". When transferred to adapt the image output, its attribution score is the gradient of activation when the output is an image of a dog. Specifically, since the last representation $\mathbf{h}$ is the input of the image generation module, we have:

$$\hat{s}_k^l = \mathbf{1} \cdot \frac{\partial \mathbf{h}}{\partial a_l^k}. \tag{15}$$

**Hyperparameter Configuration.** For our experiments, we sourced the training and testing data for the COCO dataset directly from its website[3]. Similarly, we obtained the source code for the target models GILL[4] and NExT-GPT[5], the image segmentation model EVA02[6], and the attribution algorithm Diffuser Interpreter[7] used for stable diffusion from the links cited in their respective publications

All hyperparameter settings, such as the division of training and testing datasets, learning rate, and optimizer, are consistent with the original configurations of the above link unless otherwise stated. Additionally, it is important to note that, unless explicitly mentioned, the samples used in the experiments were 500 images randomly selected from the COCO dataset.

Furthermore, we use Quadro RTX6000 GPUs with 24GB of memory as a representative example of consumer-level GPUs; 40GB A100s and 80GB H100s to provide datacenter-level benchmarks.

---

[3]COCO: http://images.cocodataset.org

[4]GILL: https://github.com/kohjingyu/gill

[5]NExT-GPT: https://github.com/NExT-GPT

[6]EVA-02: https://github.com/baaivision/EVA/

[7]Diffuser-Interpreter: https://github.com/JoaoLages/diffusers-interpret

Table 4: The semantics of T/I-neurons with the highest attribution scores identified by different attribution methods, when the output of MLLMs containing the targeted semantics. (L$l$,U$k$) denotes the $k$-th neuron at layer $l$. For each method, we report semantics with top-4 probabilities.

| Output | Segmentation | Category | Method | Location | Semantics |
|---|---|---|---|---|---|
| | | T-Neurons | Grad | L30.U10180 | {'card', 'screen','Card'} |
| | | | AcT | L26.U8761 | {'coffee', 'tea', 'brew'} |
| | | | CE | L27.U8520 | {'CLSID', 'Nanto', 'Kinnikuman'} |
| | | | **NAM** | **L27.U9810** | {**'glass', 'Glass', 'bud'**} |
| | | I-Neurons | Grad | L28.U3335 | {'spirits', 'spirit', 'run'} |
| | | | AcT | L30.U5844 | {'bit', 'lot', 'few'} |
| | | | CE | L31.U3393 | {'order', 'Order', 'orders'} |
| | | | **NAM** | **L27.U413** | {**'drink', 'drinking', 'drinks'**} |
| | | T-Neurons | Grad | L30.U4516 | {'roll', 'rolls', 'Roll'} |
| | | | AcT | L28.U6023 | {'eat', 'eating', 'eaten'} |
| | | | CE | L25.U11833 | {'flexible', 'brittle', 'bend'} |
| | | | **NAM** | **L30.U8704** | {**'taste', 'tasted', 'tasting'**} |
| | | I-Neurons | Grad | L30.U14400 | {'fruit', 'rose', 'apple'} |
| | | | AcT | L27.U6886 | {'snap', 'taken', 'snapped'} |
| | | | CE | L25.U14616 | {'Kod', 'negative', 'develops'} |
| | | | **NAM** | **L27.U2615** | {**'cookies', 'pancakes', 'baked'**} |
| | | T-Neurons | Grad | L31.U404 | {'vaults', '70710', '20439'} |
| | | | AcT | L27.U8950 | {'driver', 'derivers', 'vehicle'} |
| | | | CE | L24.U15330 | {'few', 'lot', 'bit'} |
| | | | **NAM** | **L25.U3913** | {**'bike', 'bikes', 'cycle'**} |
| | | I-Neurons | Grad | L28.U1208 | {'20439', 'Kinnikuman', 'Nanto'} |
| | | | AcT | L27.U6437 | {'gun', 'car', 'bike'} |
| | | | CE | L25.U4343 | {'CLSID', 'shapeshifter', 'couple'} |
| | | | **NAM** | **L25.U3913** | {**'bike', 'bikes', 'cycle'**} |

**Ablation Study.** Our attribution score design primarily consists of two components: activation and its mapped values on the target semantic dimension in the vocabulary. We aim to leverage the complementary effects of both components, using them as indicators of indirect and direct contributions, respectively. The results of baseline methods AcT and AcU can be approximated as the attribution effects when using each component separately. Therefore, comparing our method with AcT and AcU can also be considered as an ablation study. This comparison also reflects that using only the mapped values (AcU) can introduce certain biases, leading to poorer performance across various metrics.

**External Model & Algorithm.** Then, we introduce the semantic segmentation model EVA02 and attribution algorithm for the stable diffusion model Diffuser Interpret. Specifically, EVA02 [30] comprises a series of robustly optimized plain Vision Transformers (ViTs) of moderate model sizes, featuring transferable bidirectional visual representations learned from a powerful CLIP vision encoder via masked image modeling (MIM) pre-training. Compared to current leading vision models with billions of parameters, the EVA-02 variants necessitate significantly fewer computational resources, enabling a more in-depth exploration of often-overlooked aspects. Furthermore, Diffuser Interpret [31] attributes the pixel values of the generated image to the input embedding of the stable diffusion using gradient information. We also apply this strategy to the attribution of the projector used by GILL.

## B.2 More Experimental Results

**Results on Semantic Relevance.** To better illustrate the comparison between our attribution method and the baseline attribution methods, we present additional qualitative experimental results to provide a more intuitive comparison, as shown in Table 4. The effects demonstrated by these methods are similar to those shown in the experimental results in the main text, indicating that the neurons identified by our NAM method are semantically closer to the target semantics.

**Results on NExTGPT.** In the main body of the text, we have completed testing the effectiveness of various neuron attribution algorithms with GILL as the target model. In this section, we expand our experiments to include NExTGPT as the target model. Below, we present the key neurons identified in NExTGPT and their semantic correlations with images and captions. The results can be found in Table 5.

**Consistency Between Neuron's Semantics and Caption Given by GPT.** The main text only presents the average consistency of the T/I-neurons identified by various attribution methods with the captions of the input/output images. To refine our comparison, we provide the specific consistency

Table 5: Consistency between the neuron's semantics and the images/captions. *Grad.*, *Act.* and *Ca.* denote gradient-, activation- and causality-based methods, respectively. ‡ and ⋆ represent the CLIPScore *w.r.t* input and output images. We use background ▪ highlights the best performance.

| Method | Class | | | T-Neurons | | | | |
| | Grad. | Act. | Ca. | ‡CLipScore | ⋆CLipScore | BERTScore | MoveScore | BLEURT |
|---|---|---|---|---|---|---|---|---|
| CE | | | ✓ | $0.258_{\pm0.017}$ | $0.244_{\pm0.020}$ | $0.257_{\pm0.032}$ | $0.261_{\pm0.025}$ | $0.038_{\pm0.006}$ |
| GraI | ✓ | ✓ | | $0.228_{\pm0.018}$ | $0.233_{\pm0.022}$ | $0.268_{\pm0.034}$ | $0.290_{\pm0.035}$ | $0.042_{\pm0.006}$ |
| GraD | ✓ | | | $0.369_{\pm0.051}$ | $0.387_{\pm0.029}$ | $0.456_{\pm0.029}$ | $0.402_{\pm0.024}$ | $0.059_{\pm0.011}$ |
| GraT | ✓ | ✓ | | $0.406_{\pm0.034}$ | $0.409_{\pm0.033}$ | $0.485_{\pm0.026}$ | $0.480_{\pm0.042}$ | $0.068_{\pm0.008}$ |
| AcT | | ✓ | | $0.607_{\pm0.036}$ | $0.587_{\pm0.044}$ | $0.608_{\pm0.065}$ | $0.632_{\pm0.055}$ | $0.120_{\pm0.014}$ |
| AcU | | ✓ | | $0.593_{\pm0.048}$ | $0.615_{\pm0.053}$ | $0.592_{\pm0.046}$ | $0.611_{\pm0.048}$ | $0.141_{\pm0.020}$ |
| NAM | | ✓ | | $0.614_{\pm0.055}$ | $0.620_{\pm0.055}$ | $0.616_{\pm0.042}$ | $0.633_{\pm0.036}$ | $0.145_{\pm0.021}$ |

| Method | Class | | | I-Neurons | | | | |
| | Grad. | Act. | Ca. | ‡CLipScore | ⋆CLipScore | BERTScore | MoveScore | BLEURT |
|---|---|---|---|---|---|---|---|---|
| CE | | | ✓ | $0.217_{\pm0.024}$ | $0.210_{\pm0.019}$ | $0.238_{\pm0.036}$ | $0.254_{\pm0.039}$ | $0.041_{\pm0.005}$ |
| GraI | ✓ | ✓ | | $0.226_{\pm0.035}$ | $0.242_{\pm0.018}$ | $0.246_{\pm0.031}$ | $0.265_{\pm0.035}$ | $0.033_{\pm0.005}$ |
| GraD | ✓ | | | $0.362_{\pm0.037}$ | $0.369_{\pm0.031}$ | $0.418_{\pm0.044}$ | $0.388_{\pm0.042}$ | $0.055_{\pm0.007}$ |
| GraT | ✓ | ✓ | | $0.417_{\pm0.043}$ | $0.386_{\pm0.030}$ | $0.466_{\pm0.039}$ | $0.421_{\pm0.045}$ | $0.069_{\pm0.009}$ |
| AcT | | ✓ | | $0.533_{\pm0.042}$ | $0.564_{\pm0.048}$ | $0.568_{\pm0.062}$ | $0.608_{\pm0.073}$ | $0.119_{\pm0.015}$ |
| AcU | | ✓ | | $0.592_{\pm0.055}$ | $0.583_{\pm0.051}$ | $0.582_{\pm0.042}$ | $0.535_{\pm0.057}$ | $0.130_{\pm0.017}$ |
| NAM | | ✓ | | $0.608_{\pm0.047}$ | $0.585_{\pm0.049}$ | $0.592_{\pm0.041}$ | $0.635_{\pm0.067}$ | $0.142_{\pm0.018}$ |

scores *w.r.t* the captions generated by GPT for the output images. The detailed results are shown in Table 6 and 7. These results also demonstrate the highest consistency scores of our attribution method compared to the baselines.

# C Broader Impact

In this paper, we present a novel neuron attribution method NAM to interpret the MLLMs. Based on NAM, we reveal several intriguing neuron properties within MLLMs. These properties collectively elucidate the inner workings of neurons within MLLMs, enhancing our understanding of their capacity to process and generate multi-modal content. This approach can contribute to a wide range of applications of MLLMs, boosting the MLLMs across various downstream tasks such as knowledge editing [15, 13, 14]. We believe that the neuron properties drawn from NAM can shed light for future research on MLLM community, and inspire further exploration into understanding neurons within other pre-trained transformers.

Table 6: Consistency between the neuron's semantics within GILL and the captions given by GPT. *Grad.*, *Act.* and *Ca.* denote gradient-, activation- and causality-based methods, respectively. ‡ and ⋆ represent the CLIPScore *w.r.t* input and output images. We use background ⬛ highlights the best performance.

| Method | Class | | | T-Neurons (GILL) | | |
| --- | --- | --- | --- | --- | --- | --- |
| | Grad. | Act. | Ca. | BERTScore | MoveScore | BLEURT |
| CE | | | ✓ | $0.254_{\pm0.035}$ | $0.263_{\pm0.029}$ | $0.036_{\pm0.013}$ |
| GraI | ✓ | ✓ | | $0.262_{\pm0.038}$ | $0.278_{\pm0.028}$ | $0.040_{\pm0.008}$ |
| GraD | ✓ | | | $0.452_{\pm0.027}$ | $0.392_{\pm0.019}$ | $0.062_{\pm0.013}$ |
| GraT | ✓ | ✓ | | $0.476_{\pm0.029}$ | $0.475_{\pm0.038}$ | $0.066_{\pm0.011}$ |
| AcT | | ✓ | | $0.599_{\pm0.057}$ | $0.623_{\pm0.046}$ | $0.113_{\pm0.010}$ |
| AcU | | ✓ | | $0.584_{\pm0.043}$ | $0.615_{\pm0.055}$ | $0.134_{\pm0.020}$ |
| NAM | | ✓ | | $0.601_{\pm0.037}$ | $0.618_{\pm0.028}$ | $0.142_{\pm0.019}$ |

| Method | Class | | | I-Neurons (GILL) | | |
| --- | --- | --- | --- | --- | --- | --- |
| | Grad. | Act. | Ca. | BERTScore | MoveScore | BLEURT |
| CE | | | ✓ | $0.241_{\pm0.037}$ | $0.248_{\pm0.042}$ | $0.038_{\pm0.009}$ |
| GraI | ✓ | ✓ | | $0.237_{\pm0.033}$ | $0.267_{\pm0.032}$ | $0.027_{\pm0.004}$ |
| GraD | ✓ | | | $0.409_{\pm0.036}$ | $0.382_{\pm0.036}$ | $0.053_{\pm0.008}$ |
| GraT | ✓ | ✓ | | $0.470_{\pm0.035}$ | $0.408_{\pm0.041}$ | $0.070_{\pm0.011}$ |
| AcT | | ✓ | | $0.554_{\pm0.050}$ | $0.602_{\pm0.062}$ | $0.117_{\pm0.019}$ |
| AcU | | ✓ | | $0.571_{\pm0.037}$ | $0.528_{\pm0.062}$ | $0.124_{\pm0.014}$ |
| NAM | | ✓ | | $0.584_{\pm0.039}$ | $0.622_{\pm0.052}$ | $0.137_{\pm0.021}$ |

Table 7: Consistency between the neuron's semantics within NExTGPT and captions given by GPT. *Grad.*, *Act.* and *Ca.* denote gradient-, activation- and causality-based methods, respectively. ‡ and ⋆ represent the CLIPScore *w.r.t* input and output images. We use background ⬛ highlights the best performance.

| Method | Class | | | T-Neurons (NExTGPT) | | |
| --- | --- | --- | --- | --- | --- | --- |
| | Grad. | Act. | Ca. | BERTScore | MoveScore | BLEURT |
| CE | | | ✓ | $0.263_{\pm0.028}$ | $0.259_{\pm0.024}$ | $0.036_{\pm0.007}$ |
| GraI | ✓ | ✓ | | $0.264_{\pm0.037}$ | $0.288_{\pm0.035}$ | $0.040_{\pm0.008}$ |
| GraD | ✓ | | | $0.455_{\pm0.025}$ | $0.393_{\pm0.028}$ | $0.060_{\pm0.014}$ |
| GraT | ✓ | ✓ | | $0.482_{\pm0.024}$ | $0.472_{\pm0.039}$ | $0.070_{\pm0.010}$ |
| AcT | | ✓ | | $0.602_{\pm0.057}$ | $0.624_{\pm0.043}$ | $0.114_{\pm0.007}$ |
| AcU | | ✓ | | $0.585_{\pm0.037}$ | $0.623_{\pm0.042}$ | $0.135_{\pm0.016}$ |
| NAM | | ✓ | | $0.613_{\pm0.034}$ | $0.624_{\pm0.028}$ | $0.140_{\pm0.017}$ |

| Method | Class | | | I-Neurons (NExTGPT) | | |
| --- | --- | --- | --- | --- | --- | --- |
| | Grad. | Act. | Ca. | BERTScore | MoveScore | BLEURT |
| CE | | | ✓ | $0.234_{\pm0.029}$ | $0.243_{\pm0.028}$ | $0.037_{\pm0.007}$ |
| GraI | ✓ | ✓ | | $0.239_{\pm0.028}$ | $0.271_{\pm0.025}$ | $0.036_{\pm0.007}$ |
| GraD | ✓ | | | $0.417_{\pm0.040}$ | $0.382_{\pm0.037}$ | $0.054_{\pm0.008}$ |
| GraT | ✓ | ✓ | | $0.458_{\pm0.034}$ | $0.406_{\pm0.041}$ | $0.070_{\pm0.014}$ |
| AcT | | ✓ | | $0.570_{\pm0.055}$ | $0.597_{\pm0.059}$ | $0.114_{\pm0.008}$ |
| AcU | | ✓ | | $0.579_{\pm0.039}$ | $0.528_{\pm0.054}$ | $0.126_{\pm0.018}$ |
| NAM | | ✓ | | $0.588_{\pm0.033}$ | $0.627_{\pm0.057}$ | $0.138_{\pm0.020}$ |

