# OpenReview forum: "Towards Neuron Attributions in Multi-Modal Large Language Models"
_NeurIPS.cc/2024/Conference — NeurIPS 2024 poster_

### Official Review · Reviewer_gq3m · 2024-07-08

**Soundness:** 4
**Presentation:** 2
**Contribution:** 3
**Rating:** 8
**Confidence:** 4

**Summary:**

The paper describes a neuron attribution method for multimodal LLMs that the authors term "NAM". The method is broken down into two steps, a first step that uses image segmentation and a pretrained attribution algorithm, Diffusers-Interpret, to assign relevance scores to the final hidden state of the model. The second step involves attributing the final hidden state to neurons within the feed forward network of the base LLM.

**Strengths:**

The method seems to be effective at what it sets out to achieve, namely attributing model semantics to individual FFN neurons in the base model, and separating out the relative modality-specific semantic properties into I-Neurons and T-Neurons. The cross-sample invariance and semantic selectivity are compelling and convincing of the method's efficacy. The editing application gives an example of an especially relevant use case of the method.

**Weaknesses:**

The paper is at times difficult to read and needlessly obtuse, reflecting an unfortunate general trend in these sorts of papers. The paper would benefit from being less notationally dense and being less of a chore to read, especially since the underlying ideas are not nearly as complex as the notation suggests. Figure 2 is especially confusing and there are way too many moving parts given the relatively brief caption. There are some assumptions and potential limitations that may need to be justified (see questions for details).

**Questions:**

It is not clear how NAM performs in cases where one might want to perform attribution for semantic concepts that are not discrete objects which can be easily segmented out. A fundamental assumption here is that the semantics of the concept that one would like to explain have to be visually segmentable. I would like to see some discussion or clarification on whether the method is limited only to attribution of concrete, grounded concepts. For example, in the sentence "A school of fish is chased by a shark.", how would NAM attribute the word "chased"?

A clarifying question: An underlying assumption here also seems to be that it is the FFN neurons that are primarily the ones that predominantly encode semantic knowledge. The authors give some citations in the introduction to support this assumption. Is there some basic intuition for why this cannot be encoded in the self-attention block instead?

The authors state that one of the primary motivations of this method over others is that their neuron attribution method does not require intensive backwards-and-forwards passes. How much of a problem is this actually? It would be nice to have a comparison of computational costs, especially since this seems to be something that the authors argue.

I wonder if this method would be applicable to the more challenging problem of sound object identification and attribution in multimodal audio LLMs. For example, given an image of a spectrogram of a duck quacking and a person talking, could you use this technique (perhaps replacing some of the constitent pieces here) to identify the components of the spectrogram that correspond to each sound object? To be clear, I am not requesting an additional experiment here, but do want to hear the author's thoughts on the matter and whether there are any limitations to applying these ideas here.

**Limitations:**

See questions for potential concerns about limitations. No obvious negative societal impact.

---

> ### Author Rebuttal · Authors · 2024-08-05
>
> Dear Reviewer gq3m:
>
> Thank you very much for your comments. We sincerely appreciate the time and effort you have dedicated to reviewing our work! Below, we meticulously provide responses to each of your comments and outline the modifications made to the manuscript based on your suggestions. Hope that our responses could address your concerns!
>
> >***W1: The paper would benefit from being less notationally dense.***
>
> Thank you for your valuable feedback. Following your suggestions, we have revised our paper to improve clarity and readability. Specifically, we have streamlined the presentation by reducing notational density and simplifying definitions wherever possible (*e.g.*, Equation 4~6).
>
> Moving forward, we will be mindful of your suggestions and strive to maintain simplicity in both our descriptions and equations. We hope to contribute to curbing the trend of unnecessary complexity in academic writing.
>
> Thank you again for your thoughtful comments!
>
> >***Q1: I would like to see some discussion on whether the method is limited to attribution of concrete concepts.***
>
> Thank you for your insightful suggestion. To address your concern, we have expanded our discussion on the limitations of our method. Here are the key points:
>
> -	Firstly, our method relies on external segmentation models which can only segment concrete concepts. Hence, we acknowledge that this dependency makes our method challenging to attribute abstract concepts.
> -	Despite this limitation, we believe that **this dependency** (*i.e.*, leveraging established external models from mature fields) **can be beneficial**, especially during the early stages of developing a new domain. Our experiments support this approach, demonstrating the advantages of cross-domain collaboration. Moreover, as segmentation models continue to advance, the upper bound of our method's capabilities will improve.
> -	Finally, we find it pertinent to mention our ongoing project that builds on this work and aims to attribute abstract concepts within multimodal LLMs. Specifically, it seeks to capture commonalities in the hidden layer when LLMs output abstract concepts, thereby circumventing the limitations of segmentation models. We plan to release it soon after this paper is accepted.
>
> Furthermore, following your suggestion, we have added a subsection in the revised version, which elaborates on these points. We hope this addition enhances the rigor of the paper and provides readers with valuable insights.
>
> >***Q2: The authors give citations to support that FFN predominantly encodes semantic knowledge. Is there some basic intuition for why this cannot be encoded in the self-attention block instead?***
>
> Thank you for raising this important question. The prevailing view — or basic intuition — suggests that the self-attention block is primarily responsible for gathering global features, while the FFN extracts contextual information based on these features, concretizing complex features into stable semantic concepts. Hence, the FFN potentially stores information that requires abstraction and synthesis beyond the immediate context.
>
> Furthermore, from an experimental standpoint, several studies in recent years have supported this view. One of the most influential works is the ‘‘causal tracing’’ [1]. It employed neuron activation interventions and observed their impact on the output, achieving causal attribution of different blocks. We highly recommend reading this fascinating study for more detailed insights.
>
> Finally, if you feel it would be beneficial, we are prepared to **expand the Preliminary section in the revised version to include a more thorough discussion** of both the theoretical and experimental evidence supporting this hypothesis.
>
> Hope our response could address your concerns!
>
> [1] Locating and Editing Factual Associations in GPT. NeurIPS 2022
>
> >***Q3: It would be nice to have a comparison of computational costs, especially since the author claims that their neuron attribution method does not require time-consuming backwards-and-forward passes.***
>
> Thanks for your suggestion. Based on your feedback, we have added an analysis of computational costs in the revised version. The table below presents the average time (in seconds) required to attribute a concept. This demonstrates that avoiding the traditional backwards-and-forward pass significantly reduces computational overhead, effectively supporting our claims.
>
> |||||
> |:-:|:--:|:-:|:-:|
> |Method|Backwards|GILL|NExTGPT|
> |GraI|Yes|712.7|914.9|
> |GraD|Yes|279.3|299.7|
> |GraT|Yes|303.2|418.0|
> |NAM|No|**46.4**|**71.5**|
> |||||
>
> We hope these additional experiments provide a clearer understanding of the efficiency gains offered by our approach.
> Your input has been invaluable in enhancing the rigor and clarity of our paper. Thank you once again for your insightful feedback!
>
>
> >***Q4: I wonder if this method would be applicable to audio modalities.***
>
> Thank you for your insightful question. Our method can indeed be applied to other modalities such as audio, since the new attribution function we propose is modality-agnostic and can generalize across different types of data.
>
> To better illustrate this point, we conducted experiments with audio outputs using NExT-GPT. Here are the results:
> |||||
> |:-:|:-:|:-:|:-:|
> |Method|Specificity|Relevance|Invariance|
> |CE|0.217|0.403|0.274|
> |GraI|0.264|0.500|0.337|
> |GraD|0.263|0.509|0.355|
> |AcU|0.349|0.552|0.396|
> |NAM|**0.361**|**0.596**|**0.428**|
> |||||
>
> Due to time constraints, we were only able to complete this single set. We hope for your understanding in this matter. Additional experiments are currently underway, and we plan to include them in the paper after acceptance.
>
> *Once again, we are deeply appreciative of the time and expertise you have shared with us. Your insightful suggestions have significantly enhanced our manuscript, and we are more than happy to add clarifications to address any additional recommendations and reviews from you!*
>
> *Best,*
>
> *Authors*

---

> > ### Comment · Reviewer_gq3m · 2024-08-07
> > **Revised Score**
> >
> > I believe my concerns here are reasonably satisfied. I am especially impressed by the turnaround on the audio model analysis, which I think is quite a nice addition to the paper. Conditional on that the results in this response are effectively integrated into the main paper, I am raising my score to an 8 to reflect these additional improvements, and would support acceptance.

---

> > > ### Author Response · Authors · 2024-08-08
> > > **Gratitude to Reviewer gq3m for Feedback**
> > >
> > > Dear Reviewer gq3m:
> > >
> > > Thank you very much for your positive feedback and for raising your score. We greatly appreciate your detailed comments and suggestions, which have significantly improved our paper. **Your recognition encourages us to continue refining our work.**
> > >
> > > Your support and trust are invaluable to us. They not only validate our current efforts but also motivate us to push forward in the field of interpretable LLMs. We are committed to advancing our research and contributing meaningful insights and innovations to this rapidly evolving area.
> > >
> > > **While we recognize that interpretable LLMs are still in their early stages and that there is a long road ahead, we firmly believe that with persistent effort and dedication, we will make significant progress.**
> > >
> > > Thank you once again for your encouragement and support.
> > >
> > > Best regards,
> > >
> > > Authors

---

### Official Review · Reviewer_VfTi · 2024-07-10

**Soundness:** 3
**Presentation:** 3
**Contribution:** 3
**Rating:** 6
**Confidence:** 4

**Summary:**

This paper proposes a method to attribute the multimodal output to neurons that are influential in the generation process. The approach considers image-text multimodal LLM where 1) the base LLM's internal representation is directly used to generate text, and 2) also passed to an image generation model. These this two tiered model architecture, the proposed approach breaks down the problem into attribution for the base model and attribution for the image generation model, which can be combined to cover all neurons in the considered architecture. The proposed approach is evaluated with baselines and shown to outperform in several tests including cross sample invariance.

**Strengths:**

- S1: The proposed approach is efficient as it does not require backpropagation.
- S2: The proposed approach outperforms the baselines in several evaluation tasks including cross-sample invariance.
- S3: The evaluation shows additional examples showing the semantics of the neurons and image editing.

**Weaknesses:**

- W1: It is unclear how to get to the Eq 1. From the GILL paper, it is using a regular OPT/GPT architecture. How did you get the linear addition of the attention vector instead of the typical multihead self-attention? Also, normalizations should be applied after a self-attention block/linear layer is applied, making linear relation from an intermediate layer to the last layer / penultimate embedding not possible.
- W2. In the semantic relevance evaluation, relevant words are extracted using the same signal paths considered in the paper, while the signals can flow other ways for the baselines. Therefore, it is unclear if we can consider these words represent the semantics of the neurons identified by other methods. Moreover, the artificial neurons are known to be multisemantic, and neuron groups form a certain semantic instead of individual neurons in isolation. Thus, the metric is deemed to emphasize what the proposed approach is doing by leveraging its definition but might not be suitable for evaluating and comparing with the baselines.
- W3: The presentation can be improved in many places (e.g., Line 110, 209-210, 217-218, 260, the definition of specificity).

**Questions:**

- Q1: Are the bars in Fig 3(c) stacked or overlayed? Is the green region T \cup I - T \cap I?
- Q2: How do we know the activations of the neurons are signals. Why not lack of activations?
- Q3: Related work should be moved to the main text. What's the difference to AcU [2]? Is it the same as the activations through the residual stream in your approach (Sec 3.1.2.)?
- Q4: How do you know the neuron semantics and how they are correct? How do you know they are monosemantic? Why "dog", "shark" are wrong for L28.U4786?

Overall, I find the paper has quite a bit of room to improve. However, I think this paper attacks an important problem, and shows promising results. Although not all evaluation methods are convincing, this is common for a growing subfield. At least some of the evaluation results can be useful to particular use cases, and I find this work worth dissemination. I wouldn't be upset if someone want this paper to be more refined and required another full review round, given there are some obvious presentation issues.

**Limitations:**

They mentioned the limitations.

---

> ### Author Rebuttal · Authors · 2024-08-06
>
> Dear Reviewer VfTi:
>
> Thank you very much for your comments. We sincerely appreciate the time and effort you have dedicated to reviewing our work! Below, we meticulously provide responses to your comments and outline the optimizations made to the manuscript based on your suggestions. Hope that our responses could address your concerns!
>
> >***W1: How did you get the linear addition of the attention vector in Eq1?***
>
> Thanks for your question. As mentioned in Preliminary, vector $a$ in Eq 1 is the output of the attention block that has undergone normalization. Therefore, performing linear addition on $a$ is consistent with the structure of the transformer. Moreover, this linear addition is also supported by existing works on various downstream tasks, such as LLMs editing [1] and explanation [2]. Due to space limitations, we cannot elaborate further here, but we recommend reviewing these works for additional insights.
>
> Additionally, following your feedback, we have enhanced the Preliminary in the revised version, particularly the descriptions related to Eq1. We believe this can strengthen the rigor and readability of the theoretical parts of our manuscript.
>
> Hope our response could address your concern!
>
> [1] Locating and Editing Factual Associations in GPT. NeurIPS 2022
>
> [2] Multi-modal Neurons in Pretrained Text-Only Transformers. ICCV 2023
>
> >***W2: The semantic relevance is deemed to emphasize what the proposed approach is doing, but might not be suitable for comparing with the baselines.***
>
> Thanks for your concern. We acknowledge that using semantic relevance for comparing different methods carries potential bias, as its theoretical foundation is similar to our approach, all based on residual connections.
>
> Therefore, following your suggestion, we have revised the experimental section in the new version. Specifically, while other metrics (*e.g.*, invariance and specificity) are retained, semantic relevance is only employed to validate the theoretical soundness of our method. We believe this adjustment will enhance the rigor of our manuscript.
>
> We hope our response could address your concern!
>
> >***W3: Presentation can be improved in Line 110, 209-210, 217-218, 260 and the definition of specificity.***
>
> We apologize for the oversight that resulted in some grammatical errors in the original text. Following your feedback, we have rechecked our paper and corrected grammatical errors and unclear expressions. Here are some examples of the corrections made:
>
> -	*Line 110: To identify the contribution, we define these contribution scores as R.*
> -	*Line 209-210: The remaining results can be found in Appendix B.*
> -	*Line 217-218: For details on NAM and the baselines, please refer to Appendix A.*
> -	*Line 260：Figure 4 (a) presents the average invariance of concepts.*
> -	*Definition of specificity: Specificity is defined as the proportion of  neurons that are crucial for a single concept solely.*
>
> We believe these revisions enhance the overall quality and readability of the manuscript. Thanks once again for your valuable comment!
>
> >***Q1: Are the bars in Fig 3(c) stacked or overlayed? Is the green region T \cup I - T \cap I?***
>
> We apologize for any confusion caused. The bars in Fig 3 (c) are **overlayed**, and the green region represents T \cap I. To avoid this confusion, we have added this clarification to the reversion.
>
> >***Q2: How do we know the activations of the neurons are signals. Why not lack of activations?***
>
> Thanks for your concern. Our work aims to explore which neurons play a critical role in generating multimodal content. Intuitively, neurons that are not activated do not seem to contribute to the output. Therefore, we naturally focus on the activated neurons solely.
>
> However, as you mentioned, it is possible that non-activated neurons could contribute to the output in a subtle or indirect way. Hence,  we have included this insight in the Discussion. In the future, we will endeavor to explore this point, and we look forward to contributing to this area.
>
> We hope our response could address your query!
>
> >***Q3: Related work should be moved to the main text. Is AcU the same as the residual stream in your approach?***
>
> Thanks for your suggestion. In response, we have moved the Related Work into the main text. Regarding the comparison with AcU, our method differs in that we consider both the activations and the residual stream. In contrast, AcU only considers the latter.
>
> >***Q4: How do you know the neuron semantics and how they are correct? How do you know they are monosemantic? Why dog and shark are wrong for L28.U4786?***
>
> We apologize for any confusion caused. Here are our clarifications:
>
> 1.	**Neuron Semantics:** As mentioned in Line 240-243, the unembedding matrix and the second linear of the FFN are treated as the projection from neurons to the vocabulary words. In this case, the word with the highest probability can be regarded as the neuron semantics. We cannot guarantee that these semantics are entirely accurate, but we consider it acceptable to use them as one of the references to assist in analyzing neuron characteristics.
>
> 2.	**Polysemantic:** Many existing works have demonstrated the polysemantic nature of neurons. Our paper does not deny polysemy; instead, it aims to identify the most dominant of the multiple semantics. Thus, it does not conflict with polysemy but rather builds upon it.
>
> 3.	**L28.U4786:** If L28.U4786 is selected as the explanation for "cat" while its semantics are "dog" and "shark", the explanation method is inaccurate. This is the point conveyed in Table 1.
>
> Following your feedback, we have incorporated these clarifications into the appropriate sections to improve clarity and readability. Hope our response could address your concerns!
>
> *Once again, we are deeply appreciative of the time and expertise you have shared with us, and we are more than happy to add clarifications to address any additional recommendations and reviews from you!*
>
> *Best,*
>
> *Authors*

---

> > ### Comment · Reviewer_VfTi · 2024-08-10
> >
> > I appreciate the authors' response to the review, and that many points were clarified in the paper along with the bias and weaknesses of the metrics (although I cannot confirm the change since the new version is somehow not visible). I would further encourage the authors to clarify the assumptions mentioned in the response to Q4 and state the limitations of the method as well as the evaluation. Since my concerns were generally on the presentation, which is seemingly being improved, I'm keeping my recommendation to accept this paper despite the limitations and weaknesses which are always present in any methods.

---

> ### Author Response · Authors · 2024-08-10
> **Additional Clarification on Reviewer VfTi’s Concern**
>
> Dear Reviewer VfTi,
>
> Thank you very much for your active engagement and valuable feedback. We are pleased to hear that our previous response addressed most of your concerns, with the exception of Q4 (*''How do your know the semantics of neurons？''*). We appreciate the opportunity to provide further clarification. **Since this year’s NeurIPS does not allow submitting a new version during rebuttal, we will provide a detailed explanation here to address your concern as thoroughly as possible within the limited space.** We assure you that more detailed enhancements have been incorporated into the new version, and we sincerely hope for your understanding and trust in this process.
>
> ***1. Residual Streams***
>
> To address your question, let’s first revisit the concept of residual streams. The output of the $l$-th layer in the LLM can be computed as follows:$$\mathbf{h}^l=\mathbf{m}^l+\mathbf{h}^{l-1}+\mathbf{a}^l$$
>
> where $\mathbf{a}^l$ and $\mathbf{m}^l$ are the output of the attention and FFN block,  respectively. Specifically, we have $\mathbf{m}^l=W_{out} \mathbf{r}^l$
> where $\mathbf{r}^l$ is the vector of the intermediate layer in FFN and $W_{out}^l$ is the output embedding matrix in FFN.
>
> Based on these, the output of the final layer $\mathbf{h}^{L}$ can be recursively derived as:$$\mathbf{h}^L=\sum_{l=1}^L\mathbf{m}^l+\mathbf{h}^0+\sum_{l=1}^L\mathbf{a}^l$$
>
> It indicates that the output of any FFN module can contribute directly through linear addition to $\mathbf{h}^L$. This linearly accumulated flow of information is referred to as the residual stream.
>
> Given the complexity of the non-linear relationships within LLMs which are difficult to intuitively understand and control, many downstream tasks (*e.g.*, LLMs editing [1,2] and interpretability [3]) focus solely on the residual stream while reasonably neglecting other non-linear flows.
>
> ***2. Neuron Contributions via Residual Streams***
>
> Revisiting the structure of LLMs, $\mathbf{h}^L$ is passed through the unembedding matrix $W_u$ to obtain a probability distribution over the vocabulary $\mathbf{y}$, with the word corresponding to the highest probability being the final output:$$\mathbf{y}=W_u\mathbf{h}^L$$
>
> Since $\mathbf{m}^l$ contributes directly to the probability distribution through the residual stream, we define this contribution as $\mathbf{y}’$:$$\mathbf{y}’=W_u\mathbf{m}^l=W_uW^l_{out}\mathbf{r}^l$$
>
> Here, $W_uW^l_{out}$ can be considered as a new unembedding matrix, and the probability distribution $\mathbf{y}’$ can be decomposed into contributions from each of the $k$ hidden neurons in $\mathbf{r}^l$:$$\mathbf{y}’=W_u\sum_{i=1}^k W^l_{\text{out}i}r_i$$
>
> Thus, $W^{l}_{\text{out}i}{r}_i^l $ represents the distribution over the vocabulary for each hidden neuron. The word with the highest probability in this distribution is the one associated with that neuron. The intuition here is that **regardless of the magnitude of a neuron’s output, its contribution will always be most inclined toward its associated word**. For further details, please refer to [1].
>
> ***3. Using These Semantics as Evaluation Metrics***
>
> As you pointed out, this method of deriving semantics could have limitations due to potential bias (since it only considers information within the residual stream). Therefore, as mentioned in our previous response, we **did not** use this as an evaluation metric in the reversion. However, given the success of focusing on residual stream in many fundamental LLM studies and downstream tasks [1,2,3], we believe these semantics still hold reference value. As such, we included this as an interesting experimental observation for visualizing neuron semantics, and **we believe this visualization has the potential to offer readers valuable insights and perspectives on the behavior of neurons within MLLMs**.
>
> ***4. Other Evaluation Metrics***
>
> Although we removed this metric in the new version, **the other metrics in our paper are unaffected by the residual stream and are therefore unbiased**. The results from these metrics successfully validate the superiority of our method, ensuring the effectiveness of our experimental section.
>
> Finally, we acknowledge that explainable MLLMs are still in their infancy, and there is currently no universally accepted metric. **This work not only aims to provide a new paradigm but also actively explores usable, fair, and open-source evaluation metrics to advance this field.** We humbly ask for your understanding and hope you can recognize the improvements we have made in the new version based on your feedback.
>
> *Thank you once again for participating in the discussion! We are ready to address any additional questions from you at any time and look forward to continuing our communication with you!*
>
> *Best regards,*
>
> *Authors*
>
> [1] Locating and Editing Factual Associations in GPT. NeurIPS 2022
>
> [2] Mass-Editing Memory in a Transformer. ICLR 2023
>
> [3] Multi-modal neurons in pretrained text-only transformers. ICCV 2023

---

> > ### Author Response · Authors · 2024-08-12
> > **Gratitude to Reviewer VfTi and Respectful Seeking for Re-evaluation**
> >
> > Dear Reviewer VfTi,
> >
> > We would like to express our sincere gratitude for the valuable insights and feedback you have provided throughout the review process. As the discussion period is drawing to a close in about one day, we would really appreciate it if you could kindly raise your score considering our follow-up responses. In the following, we would like to summarize the contributions and responses of this paper again.
> >
> > **Contributions:**
> >
> > - Innovative Framework: Our work introduces a novel NAM framework which pioneers the explanation of MLLMs and ***addresses a crucial gap in MLLMs*** (Reviewers `gq3m`, `kLxz`, `VhwX`, `VfTi`).
> > - Robust Experiments: Our experimental design is rigorous and  ***provides convincing evidence and comprehensive results***  (Reviewers `gq3m`,`VhwX`).
> > - Valuable Insights: Our study ***offers interesting and instructive observations for the development of MLLMs*** (Reviewers `VhwX`, `VfTi`).
> >
> > **Responses to Your Concerns:**
> >
> > - Neurons Semantics: We have elaborated on the theoretical reasoning, cited sources, intuitive understanding, and practical applications of our approach to neuron semantics in two rounds of responses.
> > - Evaluation: We have adjusted the evaluation metrics to treat semantics as a visualization for noval insights of neuron interpretability.
> > - Validation of Linear Addition: We provided detailed derivations and justifications for the linear addition approach, confirming its correctness.
> > - Enhanced Clarity and Presentation: We have refined the presentation at several key points you have mentioned (*e.g.*, Figure 3(c), Related Work, Line 110, 209-210, 217-218, 260) to enhance readability.
> > - Neuronal Activations as Signals: We clarified the strong relevance of this choice to the issues explored in our paper, explaining why alternatives were not suitable for signaling in our context.
> >
> > We sincerely hope that our clarifications and detailed discussions address your concerns and that you might consider supporting our submission during this final phase of the review process. Should you have any further questions or need additional clarification, please do not hesitate to contact us. We are more than willing to provide further information promptly. Hope you a nice day and thanks again.
> >
> > Warm regards,
> >
> > The Authors of Submission 10306

---

> ### Author Response · Authors · 2024-08-14
> **Gratitude and Seeking for Final Review Consideration**
>
> Dear Reviewer VfTi,
>
> We are extremely grateful for the insightful comments and the recognition you have shown toward our work. Your suggestion for optimizing the paper's presentation and experiments has significantly strengthened the rigor of our approach, making our contribution even more valuable to the community.
>
> **As our current score still hovering around the borderline, we humbly ask if you might consider the possibility of raising your score. Your support at this stage is immensely important to us!** If you believe there are still areas in our paper that could be further improved, we would be more than happy to engage in any additional discussion to address any remaining concerns, particularly as the rebuttal period is about to conclude in just a few hours.
>
> We are fully committed to advancing the field of explainable LLMs and contributing meaningfully to the community. Your feedback and support are invaluable to us in achieving this goal.
>
> Thank you once again for your thoughtful engagement.
>
> Warm regards,
>
> Authors

---

### Official Review · Reviewer_VhwX · 2024-07-11

**Soundness:** 3
**Presentation:** 2
**Contribution:** 3
**Rating:** 5
**Confidence:** 3

**Summary:**

The work introduces a novel Neuron Attribution Method (NAM) tailored for MLLM. The NAM approach aims to reveal the modality-specific semantic knowledge learned by neurons within MLLMs, addressing the interpretability challenges posed by these models. The method highlights neuron properties such as cross-modal invariance and semantic sensitivity, offering insights into the inner workings of MLLMs.

**Strengths:**

1. NAM effectively differentiates between modality-specific neurons, ensuring accurate attribution of text and image outputs to the relevant neurons. This granularity aids in understanding how MLLMs process multi-modal content.

2. The observation  "..T/I-neurons identified by our NAM are specialized, showing specificity across different semantics. They are not generally sensitive to multiple semantics, confirming their targeted functionality... " and the application of NAM to image edit are interesting.

**Weaknesses:**

The NAM method relies on advanced segmentation models (like EVA02) and attribution algorithms (like Diffuser-Interpreter) tailored for specific generation modules. This dependency could limit the method's applicability and flexibility, especially when dealing with different or emerging MLLMs and generation technologies.

**Questions:**

NA

---

> ### Author Rebuttal · Authors · 2024-08-06
>
> Dear Reviewer VhwX:
>
> Thank you very much for your comments. We sincerely appreciate the time and effort you have dedicated to reviewing our work! Below, we meticulously provide responses to each of your comments and outline the optimizations made to the manuscript based on your suggestions. Hope that our responses could address your concerns!
>
>
> >***W1: The NAM method relies on external models (e.g., image segmentation models), which could limit the method's applicability and flexibility.***
>
> Thank you for raising this important concern. We acknowledge that reliance on external models may introduce limitations in terms of applicability and flexibility. However, we believe that this dependence is acceptable in the context of multimodal LLMs explainability for several reasons. Specifically,
>
> 1. **Employment of External Models in Multimodal LLMs:** Let's first consider the current landscape of multimodal LLMs (*e.g.*, EMU [1], Unified-IO 2 [2], DreamLLM [3], RPG [4], CM3Leon [5], GILL [6], and NextGPT [7]). These models often incorporate external models (*e.g.*, diffusion models) for multimodal understanding and generation. While these external models may introduce limitations in applicability and flexibility, they have not hindered the success and rapid development of multimodal LLMs. On the contrary, these mature models from well-established domains have become significant assets for multimodal research. **Hence, we believe that the benefits of utilizing external models in this domain** (*****i.e.***, multimodal research) far outweigh the potential limitations they might introduce.**
>
> 2.	**Optimal Performance Across Diverse Task Configurations:** According to our experimental results, our method achieves optimal performance across various modalities, datasets, and metrics. This impressive outcome, achieved through the integration of external models, demonstrates the substantial advantages of cross-domain collaboration. It indicates that the limitations introduced by external models (1) have a minimal impact on our method's application to mainstream downstream tasks, and (2) do not impede our model from achieving state-of-the-art results across various tasks. Moreover, another potential advantage of cross-domain collaboration is that as the external models continue to advance, the upper bounds of our method's capabilities will also improve, expanding the range of problems it can address. **Therefore, we believe that cross-domain collaboration should not be constrained by the potential limitations introduced by external models**.
>
> 3.	**Early Stage of Multimodal Explainability:** The field of multimodal LLM Explainability is still in its early stages. Leveraging well-established external models to aid initial exploration is a common practice. We believe that as this field advances, the dependency on external models will gradually decrease, and domain-specific models will emerge to further propel research. We hope to contribute meaningfully toward achieving this goal.
>
> Additionally, to address your concern, **we have emphasized the potential limitations introduced by external models in the Limitations section of the revised version, and included the above points of discussion in the Appendix section**, hoping to inspire further thought among researchers in the field of LLMs explainability.
>
> Hope that our response and the revisions made to the manuscript could address your concern!
>
>
> *Once again, we are deeply appreciative of the time and expertise you have shared with us. Your insightful suggestions have significantly enhanced the rigor of our experimental design and enriched the content of our manuscript, and we are more than happy to add clarifications to address any additional recommendations and reviews from you!*
>
> *Best,*
>
> *Authors*
>
> [1] Generative Pretraining in Multimodality. ICLR 2024
>
> [2] DreamLLM: Synergistic Multimodal Comprehension and Creation. ICLR 2024
>
> [3] Unified-IO 2: Scaling Autoregressive Multimodal Models with Vision Language Audio and Action. CVPR 2024
>
> [4] Next-GPT: Any-to-any multimodal LLM. Arxiv 2023
>
> [5] Mastering Text-to-Image Diffusion: Recaptioning, Planning, and Generating with Multimodal LLMs. ICML 2024
>
> [6] CM3Leon Scaling Autoregressive Multi-Modal Models: Pretraining and Instruction Tuning. Arxiv 2024
>
> [7] Generating Images With Multimodal Language Models. NeurIPS 2024

---

> > ### Comment · Reviewer_VhwX · 2024-08-09
> > **Thanks**
> >
> > Thanks for the response, I decide to keep the relative positive score.

---

> > > ### Author Response · Authors · 2024-08-10
> > > **Thank you for your support!**
> > >
> > > Dear Reviewer,
> > >
> > > Thank you sincerely for your valuable and positive feedback.   We deeply appreciate your recognition of our efforts in addressing the concerns from the reviews.
> > >
> > > Your acknowledgment of the improvements we have made is greatly encouraging, and we are grateful for your decision to maintain the positive score based on these enhancements.   We will continue to refine and develop our research to contribute meaningfully to the field.
> > >
> > > Thank you once again for your careful consideration and support.
> > >
> > > Best regards,
> > > Authors!

---

> ### Author Response · Authors · 2024-08-14
> **Gratitude and Seeking for Final Review Consideration**
>
> Dear Reviewer VhwX,
>
> We are extremely grateful for the insightful comments you provided and for the recognition you have shown toward our work. Your suggestion to include additional discussion on the introduction of external modules has significantly strengthened the rigor of our approach, making our contribution even more valuable to the community.
>
> **As our current score still hovering around the borderline, we humbly ask if you might consider the possibility of raising your score. Your support at this stage is immensely important to us!** If you believe there are still areas in our paper that could be further improved, we would be more than happy to engage in any additional discussion to address any remaining concerns, particularly as the rebuttal period is about to conclude in just a few hours.
>
> We are fully committed to advancing the field of explainable LLMs and contributing meaningfully to the community. Your feedback and support are invaluable to us in achieving this goal.
>
> Thank you once again for your thoughtful engagement.
>
> Warm regards,
>
> Authors

---

### Official Review · Reviewer_kLxz · 2024-07-13

**Soundness:** 2
**Presentation:** 2
**Contribution:** 2
**Rating:** 5
**Confidence:** 3

**Summary:**

Summary:
This paper introduces NAM (Neuron Attribution Method), a novel approach for attributing neurons to specific semantic concepts in multimodal large language models (MLLMs). The key contributions are: (1) A method to identify modality-specific neurons (text or image) that are crucial for particular semantic concepts. (2) Analysis of neuron properties like cross-modal invariance and semantic sensitivity. (3) A framework for multimodal knowledge editing based on the identified neurons.

The authors evaluate NAM on GILL and NExTGPT models, comparing it to several baseline attribution methods. They demonstrate NAM's effectiveness in identifying semantically relevant neurons, its cross-sample invariance, and its utility for targeted image editing tasks.

**Strengths:**

- The paper addresses an important gap in the interpretability of MLLMs, extending neuron attribution techniques from text-only models to multimodal systems.
- The authors conduct extensive experiments to validate NAM, including comparisons with multiple baselines and analysis of various neuron properties.
- The method is well-motivated, building on existing work in LLM interpretability while addressing the unique challenges of multimodal systems.
- The paper provides a detailed description of the NAM algorithm, making it potentially reproducible by other researchers.

**Weaknesses:**

- While the authors acknowledge this limitation, the evaluation is conducted on only two MLLMs (GILL and NExTGPT). Testing on a broader range of models would strengthen the generalizability claims.
- The experiments focus solely on text and image modalities. Exploring other modalities (e.g., audio, video) would provide a more comprehensive evaluation of NAM's capabilities.
- The method depends on external image segmentation models to remove noisy semantics. This introduces a potential source of bias and may limit the method's applicability to scenarios where such models are unavailable or unsuitable.
- The paper would benefit from ablation studies to isolate the impact of different components of the NAM algorithm, particularly the novel attribution score calculation.

**Questions:**

None.

**Limitations:**

Yes.

---

> ### Author Rebuttal · Authors · 2024-08-06
>
> Dear Reviewer kLxz:
>
> Thank you very much for your comments. We sincerely appreciate the time and effort you have dedicated to reviewing our work! Below, we meticulously provide responses to each of your comments and outline the modifications based on your suggestions. Hope that our responses could address your concerns!
>
> >***W1. Testing on a broader range of models would strengthen the generalizability claims.***
>
> Thank you for your insightful suggestion. Following your feedback, in addition to the GILL and NextGPT, we conducted experiments on two additional multimodal LLMs, EMU and DreamLLM. The results, summarized in the table below, demonstrate that our explanation method achieves optimal performance across all metrics, thus further validating the generalizability of our approach.
> ||||||
> |:-:|:-:|:-:|:-:|:-:|
> |Method|Model|CLipScore|BERTScore|MoveScore|
> |GraD||0.376|0.461|0.387|
> |AcT|EMU|0.469|0.582|0.595|
> |NAM||**0.499**|**0.625**|**0.634**|
> ||||||
> |GraD||0.382|0.449|0.362|
> |AcT|DreamLLM|0.510|0.502|0.568|
> |NAM||**0.538**|**0.521**|**0.593**|
> ||||||
>
> We have incorporated these additional results into the revised version of the paper, along with a more detailed analysis. Due to the limited time available during the rebuttal phase, we were able to include only these two sets of experiments. We hope for your understanding in this matter.
>
> Further experiments with additional multimodal LLMs are currently underway, and we plan to include them in the experimental section of the paper to enhance the robustness and comprehensiveness of our study. We hope that our response and the additional experiments meet your expectations.
>
> Thank you once again for your valuable feedback!
>
> >***W2. Exploring other modalities would provide a more comprehensive evaluation of NAM's capabilities.***
>
> Thanks for your valuable suggestion regarding the exploration of other modalities. Following your recommendation, we have extended our analysis beyond text and image modalities to include audio modality on NExT-GPT, which is capable of generating sound, and the following results validate its cross-modal generalizability.
> |||||
> |:-:|:-:|:-:|:-:|
> |Method|Specificity|Relevance|Invariance|
> |CE|0.217|0.403|0.274|
> |GraI|0.264|0.500|0.337|
> |GraD|0.263|0.509|0.355|
> |GraT|0.317|0.496|0.347|
> |AcT|0.346|0.561|0.406|
> |AcU|0.349|0.552|0.396|
> |NAM|**0.361**|**0.596**|**0.428**|
> |||||
>
> Due to the time-consuming nature of generating other modalities, such as video, we were unable to complete all experiments within the short rebuttal period. However, these experiments are currently in progress, and we plan to include them in the updated version of the paper once completed. We appreciate your understanding on this matter.
>
> Hope that our response and the additional experiments could meet your expectations!
>
>
> >***W3. The method depends on external models (image segmentation models), which introduces a potential source of bias.***
>
> Thanks for raising this important concern. We acknowledge that our method depends on external models, which would introduce potential bias. However, we believe that this dependency is an acceptable aspect within the study of explainability in multimodal LLMs for several reasons.
>
> 1.	**External Models in Multimodal LLMs:**
> Let's first consider the current landscape of multimodal LLMs (*e.g.*, EMU, DreamLLM, Unified-IO 2, RPG, CM3Leon, and GILL). These models rely on external models (*e.g.*, diffusion models) for multimodal generations. While these external models may introduce bias similar to the models our method employ, they have not hindered the remarkable success of multimodal LLMs. On the contrary, these well-developed models have become significant assets for multimodal research. Hence, we believe that the advantages of using external models far outweigh the potential bias introduced.
>
> 2.	**Optimal Performance Across Metrics:**
> According to our experimental results, our method has achieved optimal performance across various metrics. This indicates that the additional bias introduced by external models has a minimal impact on the model's accuracy, and does not impede our method from achieving state-of-the-art results. Moving forward, as segmentation models advance, the upper bound of our method's capabilities will continue to improve.
>
> 3.	**Early Stage of Multimodal Explainability:**
> The field of multimodal explainability is still in its early stages. Utilizing well-established external models from a mature field, such as image segmentation models, to aid initial exploration is inevitable. Furthermore, we believe that as the field advances, the dependency on external models will gradually decrease, and domain-specific models will emerge to further progress in research. We hope to contribute meaningfully toward achieving this goal.
>
> To address your concern, we have added the above discussion into the revised version to enhance the rigor of the paper.
>
> Hope our response could address your concern!
>
> >***W4. The paper would benefit from ablation studies to isolate the impact of different components.***
>
> Thank you for your concern. Following your suggestion, we have included ablation studies in the revised version, which focuses on two main components: our proposed attribution score (AS) and the denoising module (DM). Here are the results:
> ||||||
> |:-:|:-:|:-:|:-:|:-:|
> |Model|Method|Specificity|Relevance|Invariance|
> ||w/o AS|0.625|0.752|0.613|
> |GILL|w/o DM|0.660|0.759|0.635|
> ||NAM|0.674|0.782|0.644|
> ||||||
> ||w/o AS|0.598|0.643|0.527|
> |NExTGPT|w/o DM|0.611|0.661|0.532|
> ||NAM|0.624|0.689|0.546|
> ||||||
>
> These results demonstrate that each component of our design enhances the performance of our method. We hope these additional experiments could improve the overall integrity of our experiments.
>
> *Once again, we are deeply appreciative of the time and expertise you have shared with us. We are more than happy to add clarifications to address any additional recommendations and reviews from you!*
>
> *Best,*
>
> *Authors*

---

> > ### Comment · Reviewer_kLxz · 2024-08-12
> >
> > Thanks for your response and additional experiments. This addresses some of my concerns and I have raised my score.

---

> > > ### Author Response · Authors · 2024-08-12
> > > **Heartfelt Thanks for Your Encouraging Feedback**
> > >
> > > Dear Reviewer kLxz,
> > >
> > > Thank you very much for your timely feedback and for the positive adjustment to your score. We are truly delighted to see that our responses have successfully addressed your concerns. Your encouragement is incredibly valuable to our team.
> > >
> > > As you are aware, the field of explainable LLMs is still in its infancy. **Your insightful comments and suggestions throughout the review process have not only improved our work but also motivated us deeply. We are committed to continuing our research in this emerging area, striving to contribute further to its development.**
> > >
> > > Thank you once again for your essential role in refining our work and for your inspiring support.
> > >
> > > Warm regards,
> > >
> > > Authors

---

> ### Author Response · Authors · 2024-08-14
> **Gratitude and Seeking for Final Review Consideration**
>
> Dear Reviewer kLxz,
>
> We are extremely grateful for the insightful comments you provided and for the recognition you have shown toward our work. Your suggestion to include additional experiments has significantly strengthened the rigor of our approach, making our contribution even more valuable to the community.
>
> **As our current score still hovering around the borderline, we humbly ask if you might consider the possibility of raising your score. Your support at this stage is immensely important to us!** If you believe there are still areas in our paper that could be further improved, we would be more than happy to engage in any additional discussion to address any remaining concerns, particularly as the rebuttal period is about to conclude in just a few hours.
>
> We are fully committed to advancing the field of explainable LLMs and contributing meaningfully to the community. Your feedback and support are invaluable to us in achieving this goal.
>
> Thank you once again for your thoughtful engagement.
>
> Warm regards,
>
> Authors

---

### Author Rebuttal · Authors · 2024-08-06

Dear Reviewers:

We gratefully thank you for your valuable comments! We are truly encouraged by the reviewers' recognition of that our work has **addressed an important gap in MLLMs** (by all Reviewers), **provided interesting and instructive observations** (Reviewer VhwX and VfTi), and **conducted convincing and comprehensive experiments** (Reviewers kLxz and gq3m).

Here we meticulously give point-by-point responses to your comments, and further revise our manuscript following your suggestions. Hope that our responses could address all your concerns and meet the expectations of the conference committee.

Once again, we sincerely appreciate your time and effort in reviewing our paper. Your constructive criticism has been invaluable in refining our work, and **we are more than happy to add clarifications to address any additional recommendations and reviews from you**!

Best,

Authors

---

### Decision · Program_Chairs · 2024-09-25

**Decision:**

Accept (poster)

**Comment:**

This paper presents NAM, a new technique for attributing neurons to specific semantic concepts in multimodal large language models (MLLMs). The authors show NAM outperforms GILL and NExTGPT models in various tests. NAM reveals neuron properties such as cross-modal invariance and semantic sensitivity, making it useful for targeted image editing tasks.

Strength: All reviewers point out that the paper addresses a crucial point in the interpretability of MLLMs by expanding neuron attribution techniques from text-only models to multimodal systems. NAM demonstrates its effectiveness in distinguishing between modality-specific neurons and surpasses baseline methods in multiple evaluation tasks, including cross-sample invariance. The editing application showcases a particularly relevant use case for the method.

Weakness: Several reviewers point out the limitations including 1) the method depends on external powerful segmentation models and attribution algorithms tailored for specific generation modules. This dependency introduces a potential source of bias and may limit the method's applicability and flexibility to scenarios when dealing with different or emerging MLLMs and generation technologies. 2) The presentation could be improved in many places.

After careful consideration, despite the limitations, given that many of the reviewers' raised concerns have been carefully addressed in authors' revision and comments, I'm leaning toward an accept.